# Synthesizing the impacts of baseflow contribution on C-Q relationships across Australia using a Bayesian Hierarchical Model

Danlu Guo[1], Camille Minaudo[2], Anna Lintern[3], Ulrike Bende-Michl[1], Shuci Liu[1], Kefeng Zhang[4], Clément Duvert[5,6]

[1] Department of Infrastructure Engineering, University of Melbourne, Victoria, 3010, Australia
[2] EPFL, Physics of Aquatic Systems Laboratory, Margaretha Kamprad Chair, Lausanne, Switzerland
[3] Department of Civil Engineering, Monash University, Victoria, 3800, Australia
[4] Water Research Centre, School of Civil and Environmental Engineering, UNSW Sydney, NSW 2052, Australia
[5] Research Institute for the Environment and Livelihoods, Charles Darwin University, Darwin, NT, Australia
[6] National Centre for Groundwater Research and Training (NCGRT), Australia

*Correspondence to*: Danlu Guo (danlu.guo@unimelb.edu.au)

**Abstract.** Understanding concentration-discharge (C-Q) relationships can inform catchment solute and particulate export processes. Previous studies have shown that the extent to which baseflow contributes to streamflow can affect C-Q relationships in some catchments. However, the current understanding on the effects of baseflow contribution in shaping the C-Q patterns is largely derived from temperate catchments. As such, we still lack quantitative understanding of these effects across a wide range of climates (e.g., arid, tropical and subtropical). The study aims to assess how baseflow contributions, as

defined by the median and the range of daily baseflow indices within individual catchments (*BFI_m* and *BFI_range*, respectively), influence C-Q slopes across 157 catchments in Australia spanning five climate zones. This study focuses on six water quality variables: electrical conductivity (EC), total phosphorus (TP), soluble reactive phosphorus (SRP), total suspended solids (TSS), the sum of nitrate and nitrite ($NO_x$) and total nitrogen (TN). The impact of baseflow contributions is explored with a novel Bayesian hierarchical model.

For sediments and nutrient species (TSS, $NO_x$, TN and TP), we generally see largely positive C-Q slopes, which suggest a dominance of mobilisation export patterns. Further, for TSS, $NO_x$ and TP we see stronger mobilisation (steeper positive C-Q slopes) in catchments with higher values in both the *BFI_m* and *BFI_range*, as these two metrics are positively correlated for most catchments. The enhanced mobilisation in catchments with higher *BFI_m* or *BFI_range* is likely due to the more variable flow pathways that occur in catchments with higher baseflow contributions. These variable flow pathways can lead to higher

concentration gradients between low flows and high flows, where the former is generally dominated by groundwater/slow subsurface flow while the latter by surface water sources, respectively. This result highlights the crucial role of flow pathways in determining catchment exports of solutes and particulates. Our study also demonstrates the need for further studies on how the temporal variations of flow regimes and baseflow contributions influence flow pathways and the potential impacts of these flow pathways on catchment C-Q relationships.

**1 Introduction**

Understanding the causes of spatiotemporal variability in riverine chemistry is critical to support water quality management strategies for both human and environmental end-uses. The relationship between the river chemistry and streamflow (concentration-discharge, or C-Q relationship) often shows distinct patterns that are specific to water quality variables and catchments. These C-Q patterns are determined by (i) the spatial distribution of constituent sources within individual

catchments; and (ii) the interplay between the biogeochemical and hydrological processes, which controls constituent mobilisation and transport through each catchment (Ebeling et al., 2021; Godsey et al., 2019; Musolff et al., 2015). The C-Q relationship therefore tells us about the key catchment processes controlling river water quality. As such, the C-Q relationship can help inform catchment management and mitigation strategies to improve catchment water quality (Dupas et al., 2019; Moatar et al., 2020).

However, it is challenging to identify the key catchment processes from analysing C-Q relationships, due to the high variability in water quality across both space and time. First, water chemistry and streamflow characteristics can vary significantly across multiple spatial scales, from small headwater catchments (where key processes are easier to identify) (Dupas et al., 2021; Jensen et al., 2019; McGuire et al., 2014) to basin and continental scales (e.g., Dupas et al., 2019; Ebeling et al., 2021). Many previous studies have assessed the spatial variations in C-Q relationships for nutrients, carbon and geogenic water quality variables. These studies highlighted a number of critical drivers for these spatial variations, such as land use, land management, lithology and topography (e.g., Ebeling et al., 2021; Minaudo et al., 2019). Second, high-frequency water quality monitoring studies have shown high temporal variability in water chemistry (e.g., Kirchner et al., 2004; Rode et al., 2016). Besides variation in concentrations, recent high-frequency monitoring also highlighted the high variability of C-Q relationships over time. These temporal changes in C-Q relationships are driven by a series of mechanisms such as chemical build-up and flushing under varying flow magnitudes, and also by contrasting baseflow contributions during different stages of the hydrograph (Bende-Michl et al., 2013; Knapp et al., 2020; Musolff et al., 2021; Rusjan et al., 2008; Tunqui Neira et al., 2020b).

Hydrological characteristics of catchments have been highlighted as key influencing factors of the C-Q relationships of a catchment, as the catchment hydrology defines the flow pathways and magnitudes that are critical to the transport processes (Tunqui Neira et al., 2020a, 2020b). Prior studies have explored the links between C-Q relationships and baseflow index (BFI) and similar hydrological metrics at an interannual scale (e.g., Ebeling et al., 2021; Moatar et al., 2017; Musolff et al., 2015) or at the scale of storm events (e.g., Knapp et al., 2020; Minaudo et al., 2019; Musolff et al., 2021). Across both long and short timescales, a consistent finding is that, within a particular catchment, the C-Q relationship (and thus export behaviour) is dependent on whether streamflow is dominated by baseflow or quickflow, i.e., the baseflow contribution to total flow (Gorski & Zimmer, 2021; Knapp et al., 2020; Minaudo et al., 2019). These studies also identified baseflow contribution as a key driver of the variation in C-Q relationships across catchments (Musolff et al., 2015; Moatar et al., 2017). For example, Knapp et al. (2020) found that for solutes that are partly derived from atmospheric inputs, such as nitrate and chloride, mobilisation behaviours (i.e., positive C-Q slopes) often occur during events with drier antecedent conditions. For nitrate, baseflow contributions can further affect the C-Q relationships via changing the connectivity between surface flow and groundwater (Minaudo et al., 2019). Baseflow variation also affects the capacity of nutrient removal via changing the relative importance of hydrological and biogeochemical processes (Moatar et al., 2017). Further, the variation in the baseflow contribution of a catchment is also a key feature that can be linked to the shift between different dominant flow paths during low- and high-flow (e.g., von Freyberg et al., 2018), leading to contrasting sources and mobilisation behaviours for solutes and particulates. Although a substantial body of knowledge has been established on the impact of baseflow contributions on C-Q relationships, the existing studies have largely focused on catchments in temperate climates in Europe and North America (Knapp et al., 2020; Gorski & Zimmer, 2021; Minaudo et al., 2019; Musolff et al., 2015). The narrow range of climate conditions explored so far implies a potential limitation in transferring and systematically comparing new findings to other climate zones and other

parts of the world, because climate is proven a key control of the hydrological regime, especially regarding the baseflow contribution and flow paths of individual catchments (Beck et al., 2013; von Freyberg et al., 2018).

The current knowledge gap in understanding catchment export regimes for regions other than Europe and North America was partially addressed in Lintern et al. (2021), which focused on differences in water quality status and the C-Q relationships across different climate zones in the Australian continent. One remaining question that Lintern et al. (2021) highlighted is our lack of understanding of the substantial variations in C-Q relationships within each climate zone.

This study aims to assess the impact of catchment baseflow contributions on the C-Q relationships of sediment, nutrients and electrical conductivity across catchments within different climate zones in Australia. Our research questions are:

1. How well can we explain the spatial variation in C-Q slopes across Australian catchments with baseflow contributions?
2. How do baseflow contributions influence C-Q slopes within and across climate zones?

For the first question, we hypothesise that a substantial proportion of spatial variation in C-Q slopes in Australian catchment can be explained by baseflow contribution. Based on the above-mentioned literature, we hypothesise that the median baseflow contribution of a catchment is a key control of the C-Q slope of that catchment. We also hypothesise that the range of variation in the baseflow contributions of a catchment is a key control of the C-Q slope of that catchment, because a high (low) range of variation likely reflects the diversity (uniformity) of flow pathways contributing to streamflow, which may influence the activation and mobilisation of different chemical species.

Since climate strongly influences the hydrological regimes of catchments, our hypothesis for the second research question is that baseflow contributions will affect C-Q relationships differently in different climate zones. We answer our research questions and test our hypotheses with a Bayesian hierarchical model (BHM) (Gelman et al., 2013), which is an integrated framework that enables sharing information across catchments to strengthen the statistical power of explaining variation in individual catchments. The model is a powerful approach to capture water quality variability across catchments of varying conditions and record lengths, which is the case for Australian water quality data (Guo et al., 2019, 2020; Liu et al., 2021). We use a subset of the grand dataset that Lintern et al. (2021) used, which enables us to focus on representative catchments with water quality records captured under a wider range of flow conditions. As such, by analysing the impacts of baseflow contributions on C-Q relationships, this study will i) explain the variations in C-Q relationships within individual climate zones; ii) broaden the existing knowledge of how baseflow contribution impacts C-Q relationships to a wider range of climate conditions, and thus infer key constituent transport pathways in different climate zones.

## 2 Method

### 2.1 Data and study catchments

### 2.1.1 Water quality and flow data

This study relies on water quality and streamflow data collected across Australia by seven state agencies. These include: the Department of Land, Water and Planning (VIC DELWP, Victoria); WaterNSW (New South Wales); Department of Resources and Department of Environment and Science (QLD DNRME, Queensland); Department for Water and Environment (SA DEW, South Australia); Department of Water and Environmental Regulation (WA DER, Western Australia); Department of Primary Industries, Parks, Water and Environment (TAS DPIPWE, Tasmania) and Department of Environment, Parks and Water Security (NT DEPWS, Northern Territory).

All available water quality data were obtained from all seven state agencies in late 2019 and collated into a single national-scale database (see more details in Lintern et al., 2021). Quality control of the data was performed using quality codes, flags and detection limits provided by individual state agencies (as detailed in Table S1, Supplementary Materials). The dataset consists of a mixture of grab samples and high-frequency (continuously measured) water quality data. A daily average is taken if more than one water quality sample was collected on any day at any site – see the percentage of records where more than one samples were taken in one day individual catchments in Table S2 (Supplementary Materials). This is because that streamflow in Australia is largely recorded at a daily timestep, which limits our ability to analyse all high-frequency water quality. This study focuses on six water quality variables: total suspended solids (TSS), total phosphorus (TP), soluble reactive phosphorus (SRP), total nitrogen (TN), the sum of nitrate and nitrite ($NO_x$) and electrical conductivity (EC). These six variables have been included because they are of key concern for Australian riverine water quality and are well monitored across Australia both spatially and temporally, as illustrated in Lintern et al. (2021).

For each monitoring site for the abovementioned six variables, we also obtained the corresponding available daily streamflow data from the same seven state agencies as listed above. At each site, any missing or erroneous data were identified by the quality code (Table S1, Supplementary Materials) and removed for subsequent analyses. The daily streamflow data generally had good quality, with a median of < 5% missing or erroneous data for individual water quality variables across individual monitoring sites (Table S3, Supplementary Materials). These gaps and low-quality samples in the daily streamflow records were then filled in using streamflow modelled by the Australian Bureau of Meteorology (BoM)'s operational landscape water balance model (AWRA-L), which simulates daily streamflow across Australia (Frost et al., 2016).

For this study, we focused only on monitoring sites (catchments) with water quality and flow data that satisfy the following criteria:

1) Having over 50 pairs of corresponding concentration and flow data points; this ensures that the C-Q relationships observed are more robust against outliers (Lintern et al., 2021).

2) Having water quality time-series that span at least 3 years; this ensures that a wide range of water quality and flow conditions are captured (e.g., across different seasons, high and low flows).

3) At least 75% of the range of flow quantiles (with unconstrained bounds e.g., 5 to 80%, 10 to 85%) is covered by water quality samples; this ensures that C-Q relationships are not biased by samples obtained at only high or low flows for individual catchments.

We performed the above catchment selection for each water quality variable, and found a total of 157 sites (catchments). As the monitored water quality variables vary between catchments, there are 50-83 catchments used to investigate each variable. These catchments are distributed across five main Australian climate zones as defined by Lintern et al. (2021): arid, Mediterranean, temperate, subtropical and tropical (Figure 1). A summary of the temporal coverage of water quality and flow data is provided in Figure S1 in the Supplementary Materials. Water quality data generally cover the full range of flow quantiles of individual catchments (Figure S2, Supplementary Materials). Some sites are biased towards high flows, which is likely due to i) monitoring priority for high flow events to better represent export loads; ii) practical constraints to sample low flows in intermittent rivers and ephemeral streams.

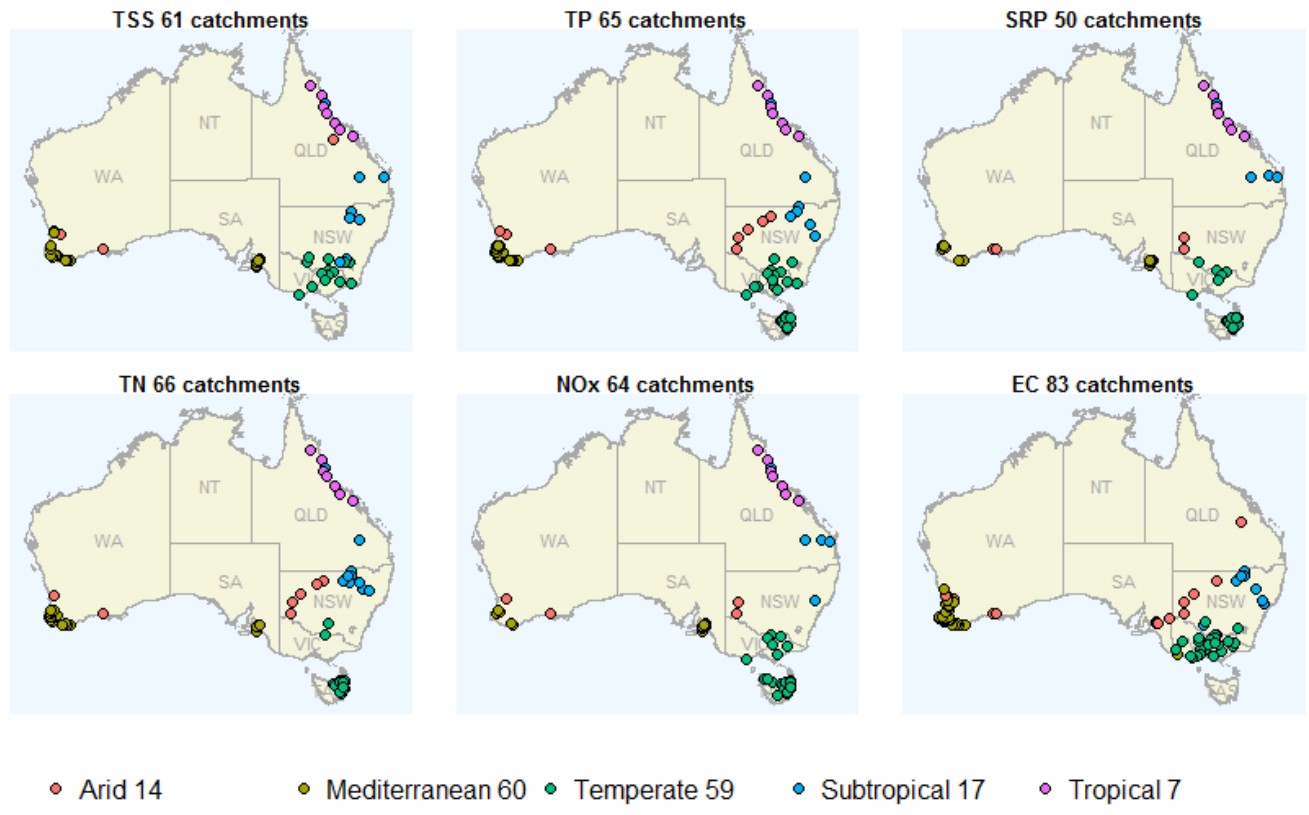

Figure 1. Catchments included in the study for each water quality variable (total number of catchments shown in panel titles). The colours denote five key climate zones in Australia. States and territories of Australia on the map are: New South Wales - NSW,

Queensland - QLD, South Australia - SA, Tasmania - TAS, Victoria -VIC, Western Australia -WA, and Northern Territory - NT. The number of catchments across all six water quality variables for each climate zone is specified in the legend.

### 2.1.2 Representing catchment baseflow contribution with baseflow index

We summarise catchment baseflow contribution with the baseflow index (BFI), which represents the proportion of discharge that occurs as baseflow (Eckhardt, 2008; Lyne & Hollick, 1979; Nathan & McMahon, 1990; Zhang et al., 2017). The daily BFIs were estimated using a Lynne-Hollick baseflow filter with Alpha = 0.98 and a burn-in period of 30 days at both ends of the time series, as recommended for the Murray-Darling Basin in south-eastern Australia (Ladson et al., 2013), within which a large number of the study catchments are located.

We aim to test our hypothesis that the median and the range of variation in catchment baseflow contributions are key controls of the C-Q slopes, based on previous literature. Therefore, for each of the 157 catchments we used two metrics of the daily BFI, namely *BFI_m* and *BFI_range*. *BFI_m* takes the median of all daily BFIs, which represents the overall baseflow contribution of the catchment. *BFI_range* is the difference between the $10^{th}$ and $90^{th}$ percentiles of daily BFIs (*BFI_10^{th}* and *BFI_90^{th}*). These quantile-based metrics were preferred over the mean and standard deviation as they are more robust against
outliers.

### 2.2 Modelling the impacts of catchment baseflow contribution on C-Q slopes

We developed a Bayesian hierarchical model (BHM) to explore the effect of catchment baseflow contributions on C-Q slopes. The key reason for choosing this model is the high heterogeneity in the national C-Q dataset in both the record period and the representation of individual climate zones, as illustrated in Section 2.1.1. BHM is effective in handling data-limited situations
via its 'information sharing' or 'borrowing power' across space (Gelman et al., 2013; Webb & King, 2009), which has been shown to be highly effective in explaining variability in spatial-temporal data under data-limited situations. This has been highlighted in several recent studies in modelling water quality over large regions in Australia (Guo et al., 2019, 2020; Liu et al., 2021). Another advantage of BHM is the ability to account for uncertainty, which is especially important for analysing water quality data, as these data are often associated with high uncertainty due to sparse sampling of the natural variability of
chemical species in river flow (Guo et al., 2020; Liu et al., 2021).

The model considered a classic C-Q relationship for any site *s* at any time-step *t* (Eqn. 1), where $\beta_s$ specifies the C-Q slope for a catchment (Godsey et al., 2009):

$$log\left(C_{s,t}\right) = \alpha_s + \beta_s log\left(Q_{s,t}\right) \tag{1}$$

Our model is based on such a single C-Q relationship at each catchment. However, our model enables the slope term ($\beta_s$) for
individual catchments to change according to their baseflow contributions. This model conceptualisation is based on previous literature on the effects of baseflow contribution on C-Q slopes within individual catchments (Gorski & Zimmer, 2021; Minaudo et al., 2019), while aiming to further explore the impact of baseflow contributions on C-Q slopes across multiple

catchments. We assume that for each water quality variable, the C-Q slopes of all catchments have a 'grand mean', $\beta_0$. Then the variation of C-Q slopes between catchments, away from $\beta_0$, is explained by changes in the catchment baseflow contribution.

The use of a mean C-Q slope here is based on our preceding study across Australian catchments, which suggested that for each water quality variable, export patterns – as represented by C-Q slopes – did not differ between climate zones (Lintern et al., 2021). Our model conceptualisation also assumes that the catchment baseflow contribution is the only controlling variable of the spatial variation of C-Q slopes, enabling us to understand how well the C-Q slopes can be explained solely by differences in baseflow contributions across catchments. We chose to investigate the effects of baseflow contributions for individual

climate zones separately to identify any statistically significant differences of these impacts between climate zones. If there is no significant difference between climate zones, the model is also capable of indicating this – as would be shown with similar, undistinguishable modelled effects of baseflow contribution for individual climate zones. Thus, our BHM incorporates different models for individual climate zones and compares them within one comprehensive modelling framework.

Thus, the resultant catchment C-Q slope $\beta_s$ is:

$$\beta_s = \beta_0 + BFI_s \times \delta_{BFI\_climate} \tag{2}$$

In Eqn. 2, $BFI_s$ is a catchment-scale metric of the baseflow contribution, i.e., $BFI\_m$ or $BFI\_range$. For each of these two metrics, its effect on the C-Q slope is considered with the climate-specific model parameter $\delta_{BFI\_climate}$ to assess whether the effects of baseflow contribution differ between climate zones.

The model conceptualisation is illustrated in Figure 2 with daily flow time series from two catchments (panel a) and the time-

series of daily BFIs along with its median ($BFI\_m$) and the 10th and 90th percentiles used to calculate $BFI\_range$ (panel b). Figure 2c illustrates the modelled C-Q slope for EC for the two catchments, $\beta_1$ and $\beta_2$, for which $BFI\_m$ was considered as the main predictor following Eqn. 2. The alternative model structure with $BFI\_range$ as the main predictor of C-Q slope was developed following the same rationale.

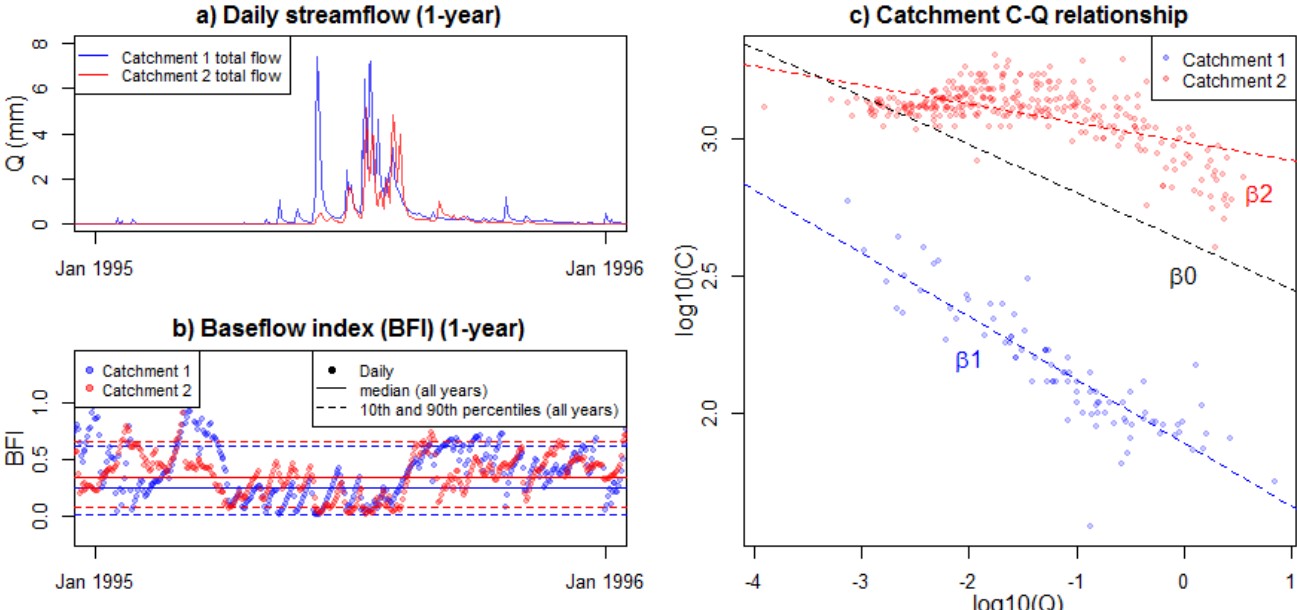

**Figure 2. Illustration of conceptualisation of the BFI-based C-Q models (Eqn. 2) with the flow and EC data from two catchments. The catchment median BFI (*BFI_m*) is used as the main predictor of C-Q slope. a) daily flow time-series; b) daily BFI time-series and the corresponding median (*BFI_m*) and the 10th and 90th percentiles. c) C-Q relationships for the two catchments, where the shift in C-Q slope ($\beta_1$, $\beta_2$) away from the grand mean $\beta_0$ is determined by *BFI_m*. Both time-series for the daily flow (a) and BFI (b) are only shown for one year for visualisation.**

Our conceptualisation of the effects of baseflow contribution leads to a modified C-Q relationship for each catchment as:

$$log\left(C_{s,t}\right) = \alpha_s + (\beta_0 + BFI_s \times \delta_{BFI\_climate}) \times log\left(Q_{s,t}\right) \qquad (3)$$

Equation 3 is the final form of the BHM, which was calibrated for each water quality variable across all catchments simultaneously. *BFI_m* and *BFI_range* were each used in separate models to independently assess the effects of these two metrics on C-Q slopes.

To calibrate the BHM, we used the R package *rstan* (Stan Development Team, 2018). The package first sampled parameter values from the Bayesian prior distributions with Markov chain Monte Carlo, and then evaluated candidate models to derive the posterior parameter distributions. Each of the unknown model parameters, $\beta_0$, $\alpha_s$ and $\delta_{BFI\_climate}$, was independently derived by sampling from a minimally informative normal prior distribution of *N(0,10)* (Gelman et al., 2013; Stan Development Team, 2018). We used four independent Markov chains in each model run, with a total of 50,000 model iterations for each chain.

Convergence of the chains was ensured by checking the *Rhat* value (Sturtz et al., 2005), which is the *rstan* output that summarises the consistency of the four Markov chains used in model calibration. Specifically, we ensured that the *Rhat* value is below 1.1, which suggests that the independent Markov chains have been well mixed and converged (Stan Development

Team, 2018). The stan codes for both models (with either *BFI_m* or *BFI_range* as the main predictor) are included in Figures S10-11 Supplementary Information.

To interpret the calibrated models, we focused on the performance and on the model parameter $\delta_{BFI\_climate}$, which informs the climate-specific effects of catchment baseflow contributions on C-Q slopes. We specifically assessed the following model outputs as presented in Section 3.3:

1) *Model performance:* For each BFI-based model of the C-Q slopes (Eqn. 3 with either *BFI_m* or *BFI_range* as the main predictor), we assessed the model performance with the $R^2$ calculated between the observed and simulated

catchment C-Q slopes, which quantifies the proportion of variance in C-Q slopes that is explained by our model. As a benchmark, we also assessed the $R^2$ of a baseline model which only allows a single parameter for each baseflow metric (*BFI_m* and *BFI_range*) across all climate zones. Comparison of our climate-specific model with this benchmark model enabled us to quantify the benefit of considering climate-specific effects of baseflow contribution on the C-Q relationships.

2) *Modelled effects:* We extracted the direction, magnitude and significance of the model parameter $\delta_{BFI\_climate}$ from the posterior distribution of the calibrated model, to infer the impact of baseflow contribution for each climate zone.

## 3 Results and Discussion

In this section, we first discuss the spatial variation in *BFI_m* and *BFI_range* across the study catchments (Section 3.1). We then provide some examples from specific catchments to illustrate how baseflow contribution can affect C-Q relationships as

a proof of concept (Section 3.2). Section 3.3 then presents the inferences made with the BFI-based C-Q model, focusing on the modelled effects of catchment baseflow contribution on C-Q slopes.

### 3.1 Baseflow contribution across catchments

The range of *BFI_m, BFI_10th* and *BFI_90th* for all catchments included in this study are summarised in Figure 3a). The calculated median BFIs are consistent with previous studies of BFI patterns in Australian catchments (Zhang et al., 2017) and

245 do not seem to correlate with catchment area (Figure S3, Supplementary Material). Generally, temperate catchments have the highest *BFI_m*, while similar *BFI_m* values are seen across the other four climate zones. *BFI_10th* and *BFI_90th* have distributions consistent with *BFI_m* in all climate zones. As different catchments were analysed for each water quality variable, the same BFI metrics were also generated for each water quality variable, and their distributions are generally consistent across different variables (Figure S4, Supplementary Materials).

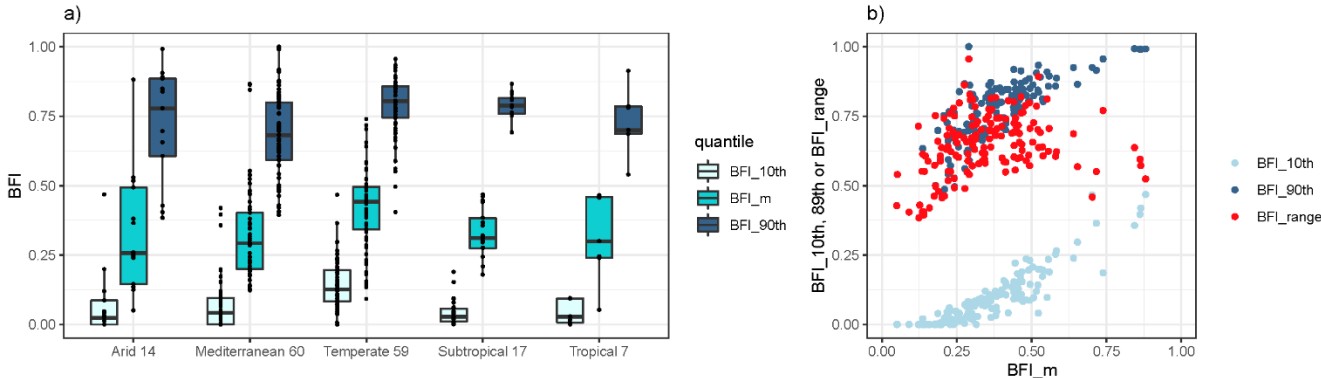

Figure 3. a) Distribution of catchment median, and the 10th and 90th percentiles of daily BFI (*BFI_m, BFI_10th, BFI_90th*) for each climate zone along with the number of catchments analysed (x-axis); b) the 10th and 90th percentiles of daily BFI (*BFI_10th* and *BFI_90th*), and *BFI_range* (*BFI_90th* – *BFI_10th*) versus *BFI_m*. Both plots include all 157 catchments across the six water quality variables studied. The corresponding plots for catchments analysed in individual water quality variables are in Figures S4 and S5 255 in the Supplementary Materials.

In general, catchments with high median BFI are likely to have a greater range of variation of daily BFI, as highlighted by the generally increasing *BFI_range* with higher *BFI_m* (Figure 3b), Spearman's $\rho$ = 0.33). The link between *BFI_m* and *BFI_range* suggests that catchments with higher *BFI_m* values are more likely driven by highly variable flow pathways. Specifically, a catchment with a low *BFI_m* tends to be associated with a small range of daily BFI (low *BFI_range*); thus, the

260 catchment is likely to always have constantly low contributions of baseflow and higher contributions of quickflow, during both dry and wet conditions. In contrast, a catchment with a high *BFI_m* generally has a large range of daily BFIs (high *BFI_range*). This means that the catchment is more likely to switch between groundwater contributions in dry conditions (high daily BFI) and surface water contributions during wet conditions (low daily BFI). However, we also note that a small proportion of catchments (9 catchments) with the highest *BFI_m* (>0.6) actually have smaller *BFI_range* compared to other catchments with

265 mid-range BFI values (0.4-0.6). This is a result of *BFI_10th* and *BFI_90th* both increasing with *BFI_m*, while the increase in *BFI_90th* plateaus at high *BFI_m*. This nonlinearity suggests that the full distribution of catchment baseflow contributions might not be sufficiently represented by either the *BFI_m* or *BFI_range* alone, providing further justification for the need to explicitly consider both the overall condition and the variation in catchment baseflow contributions when studying their effects on C-Q relationships.

**3.2 Impact of baseflow contribution on C-Q slope: proof of concept**

Before presenting the modelled effects of catchment baseflow contribution on C-Q relationships, we show some examples of individual catchments to illustrate how C-Q relationships vary across catchments with *BFI_m* and *BFI_range*. We focus on the C-Q relationships of TSS for four catchments including two arid catchments (ARIDa, ARIDb) and two tropical catchments (TROPa, TROPb) (Figure 4). For each climate zone, we include one catchment with low *BFI_m* (ARIDa, TROPa) and another

275 one with high *BFI_m* (ARIDb, TROPb), relative to the corresponding range of *BFI_m* for TSS (Figure S4).

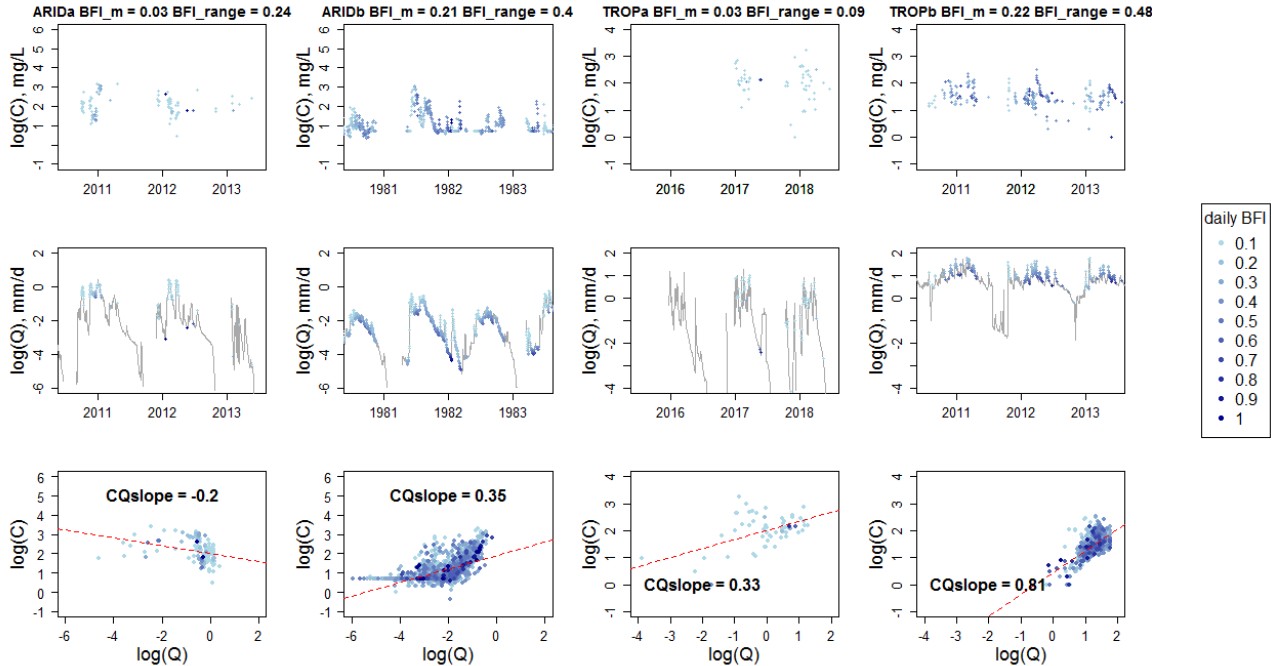

**Figure 4. C-Q relationships between TSS and flow for four individual catchments (in columns), including two arid catchments (ARIDa, ARIDb), and two tropical catchments (TROPa, TROPb). Within each climate, two catchments with a low and high value of *BFI_m* are included. The *BFI_m* along with the corresponding *BFI_range* values for individual catchments are shown in the column titles. The top and middle rows for each catchment show a 3-year time series within the records of TSS concentrations and the continuous records of flow, with dots showing the timesteps of water quality samples. The bottom row shows the C-Q relationship with all matching TSS and flow data at each catchment. All data points are coloured according to the daily BFI and all C-Q values are plotted in log-10 scale. The red dashed lines represent the observed C-Q relationship.**

Due to the particulate nature of TSS, we would expect the C-Q relationship to show a strong mobilisation behaviour that is enhanced during storm events (Musolff et al., 2015). However, our results show this is not always the case (Figure 4). For the arid catchments, the low-BFI catchment is largely dominated by quickflow (ARIDa, *BFI_m* = 0.03, *BFI_range* = 0.24, most daily BFIs are around 0.1), which has a negative C-Q slope. In contrast, the high-BFI catchment shows a non-linear C-Q relationship (in log-log space) spreading across a much wider range of daily BFIs (ARIDb, *BFI_m* = 0.21, *BFI_range* = 0.4). The overall C-Q slope for ARIDb is positive, which however, consists of a negative slope for lower flows, followed by a positive slope when the flow passes a certain threshold (around log10(Q) = -2). This is similar to the differences in C-Q slopes across low/high flows as seen in previous studies (e.g., Moatar et al., 2017), and suggests that the mobilisation behaviour for TSS occurs only during high-flow events (Thompson et al., 2011). A possible explanation for the negative C-Q slope for TSS during low flows is the dominance of biogeochemical processes rather than hydrological processes (Moatar et al., 2017).

Both tropical catchments (TROPa and TROPb) exhibit positive C-Q slopes that are relatively linear (in log-log space), where seasonal patterns in TSS are in phase with those of streamflow. Similar to the patterns seen in the two arid catchments, the catchment with higher *BFI_m* (TROPb) is associated with a wider range of daily BFI values (*BFI_m* = 0.22, *BFI_range* =

0.48). This highlights consistently strong mobilisation behaviours, with a more positive C-Q slope for the catchment that has higher $BFI\_m$ and $BFI\_range$. Overall, this preliminary analysis on a small subset of catchments suggests that baseflow contribution may indeed drive differences in C-Q relationships between catchments, and that these effects may vary across climate zones. However, it is difficult to conclude on the individual impact of $BFI\_m$ and $BFI\_range$ on the C-Q slopes, from these individual examples. The separate impacts of the two metrics are evaluated over a wide range of catchment conditions across the Australian continent with the model outputs from our BHM (Section 3.3).

### 3.3 Modelled effects of baseflow contribution on C-Q slopes

Using catchment-level metrics of baseflow contribution alone (either $BFI\_m$ or $BFI\_range$) can explain up to 22% of the variation in catchment C-Q slopes. Although these results represent limited model predictive capacity, the model does cover a large range of catchment conditions such as contrasting land uses and hydro-climate conditions. Therefore, the amount of variation that can be explained by a single BFI metric highlights baseflow contribution as an important factor that influences catchment C-Q relationships. Further, it is also worth highlighting that incorporating climate-specific impacts of baseflow contribution is highly beneficial in explaining these variations. For all six water quality parameters, the baseline model – which uses a lumped effect of catchment baseflow contribution across different climate zones – can barely explain any variation in the C-Q slopes (with all $R^2 < 0.08$, i.e., <8% of the variation explained). In contrast, the climate-specific models generally offer up to 20% increase in the variance explained for C-Q slopes, except for EC and SRP, for which performance is equally low regardless of whether the effects of baseflow contribution are separated for individual climates. The low performances for EC and SRP are likely attributed to the smaller magnitudes of C-Q slopes as highlighted in the lower median C-Q slope in Table 1, making it statistically more difficult to explain variations across catchments for these two water quality variables. These results further emphasise that in general, the impacts of catchment baseflow contribution on C-Q slopes are better defined within individual climate zones, which confirms the validity of our BFI-based C-Q models (Eqn. 3).

**Table 1. Performance of the BFI-based C-Q models – the columns show four alternative model structures with $BFI\_m$ or $BFI\_range$ as the key predictor, and with the impacts of baseflow contribution considered as lumped or specific to individual climate zones. The rows show results for individual water quality parameters. All model performances are summarised by $R^2$, which quantifies the percentage of variance in C-Q slopes explained by the BFI-based models.**

| WQ parameter | Median C-Q slope | Current (climate-specific impacts) | | Baseline (lumped impact across climate zones) | |
|---|---|---|---|---|---|
| | | *BFI_m* | *BFI_range* | *BFI_m* | *BFI_range* |
| TSS | 0.15 | 0.16 | 0.11 | 0 | 0.04 |
| TP | 0.09 | 0.14 | 0.17 | 0 | 0.08 |
| SRP | 0.06 | 0.02 | 0 | 0.03 | 0.05 |
| TN | 0.09 | 0.18 | 0.12 | 0.02 | 0.03 |
| NO$_x$ | 0.36 | 0.22 | 0.18 | 0.03 | 0 |

| EC | -0.07 | 0 | 0.01 | 0 | 0.01 |

Our climate-specific, BFI-based C-Q models synthesised the patterns observed in individual catchments (as illustrated in Section 3.2) across the Australian continent. The models suggest that both $BFI\_m$ and $BFI\_range$ have a significant influence on the C-Q slope for most climate zones and water quality parameters, and that these influences differ between climate zones

for each parameter. Figure 5 presents the median and the 95% credible intervals of these modelled effects for each water quality variable. The 95% credible interval is the range between the 2.5th and 97.5th percentiles of the posterior distribution of the parameter values, which was derived from the Bayesian posterior estimates of $\delta_{BFI\_climate}$ (Eqn. 3) to quantify the uncertainty in the modelled effects (Gelman et al., 2013). The effects of $BFI\_m$ and $BFI\_range$ on the C-Q slopes are almost always

significant, with the 95% credible intervals not crossing over 0 for most combinations of water quality variables and climate zones. An exception is for SRP, for which $BFI\_range$ always has a non-significant effect on the C-Q slopes.

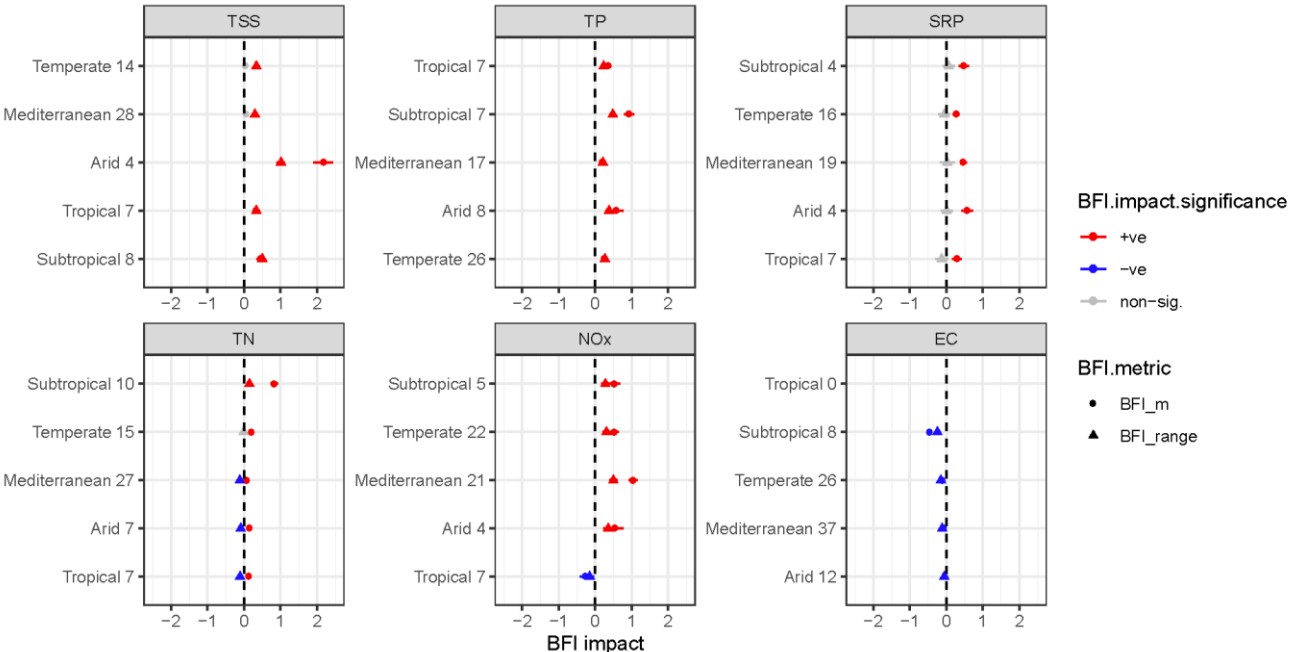

**Figure 5. Modelled effects of *BFI_m* and *BFI_range* on catchment C-Q slopes for each climate zone ($\delta_{BFI\_climate}$) for each water quality parameter. The bars show the 95% credible intervals (the range between 2.5th to 97.5th percentiles of Bayesian posterior**
**distribution) of the modelled effects, and the dots indicate the corresponding median levels. The colours indicate whether an effect is significantly positive (red), significantly negative (blue), or non-significant (grey); a positive effect means that the C-Q slope increases with a higher catchment *BFI_m* or *BFI_range*, and vice versa. Black dashed lines show the zero-effect i.e., no effect at all. The plot includes results from models with each of *BFI_m* and *BFI_range* as the key predictor, which are differentiated by marker shapes.**

To put the impacts of baseflow contribution shown in Figure 5 into context, we present the modelled catchment C-Q slopes against the corresponding $BFI\_m$ and $BFI\_range$ values in Figure 6a and b, respectively. Sediment and nutrients are largely

dominated by mobilisation, as evidenced by the large proportion of positive C-Q slopes for TSS, TP, SRP, TN and $NO_x$. In contrast, salts (EC) have largely negative C-Q slopes and are thus dominated by dilution. Regarding the effects of catchment baseflow contribution, we first note a common overall pattern for both *BFI_m* and *BFI_range*: for each water quality variable,

the fitted relationships between C-Q slopes and each BFI metric (either *BFI_m* or *BFI_range*) have a consistent 'diverging' pattern between climate zones. This is a result of our model structure, in which, for each water quality variable, all catchment C-Q slopes share a common 'grand mean' (Section 2.2) – representing a stable export pattern across Australian climate zones that is specific to the water quality variable (Lintern et al., 2021). The deviation of slopes within each climate zone from the 'grand mean' is dependent on catchment *BFI_m* or *BFI_range* (Eqn. 2). Therefore, for catchments with low *BFI_m* (or low

*BFI_range*), the differences in C-Q slopes between climate zones are smaller, and are all close to the 'grand mean'. Conversely, the C-Q slopes of catchments with high *BFI_m* (*BFI_range*) are affected more strongly by the differences between climate zones. Since these diverging patterns are a result of the model structure, we do not relate these patterns further to any physical interpretation on the impacts of BFI metrics on C-Q slopes.

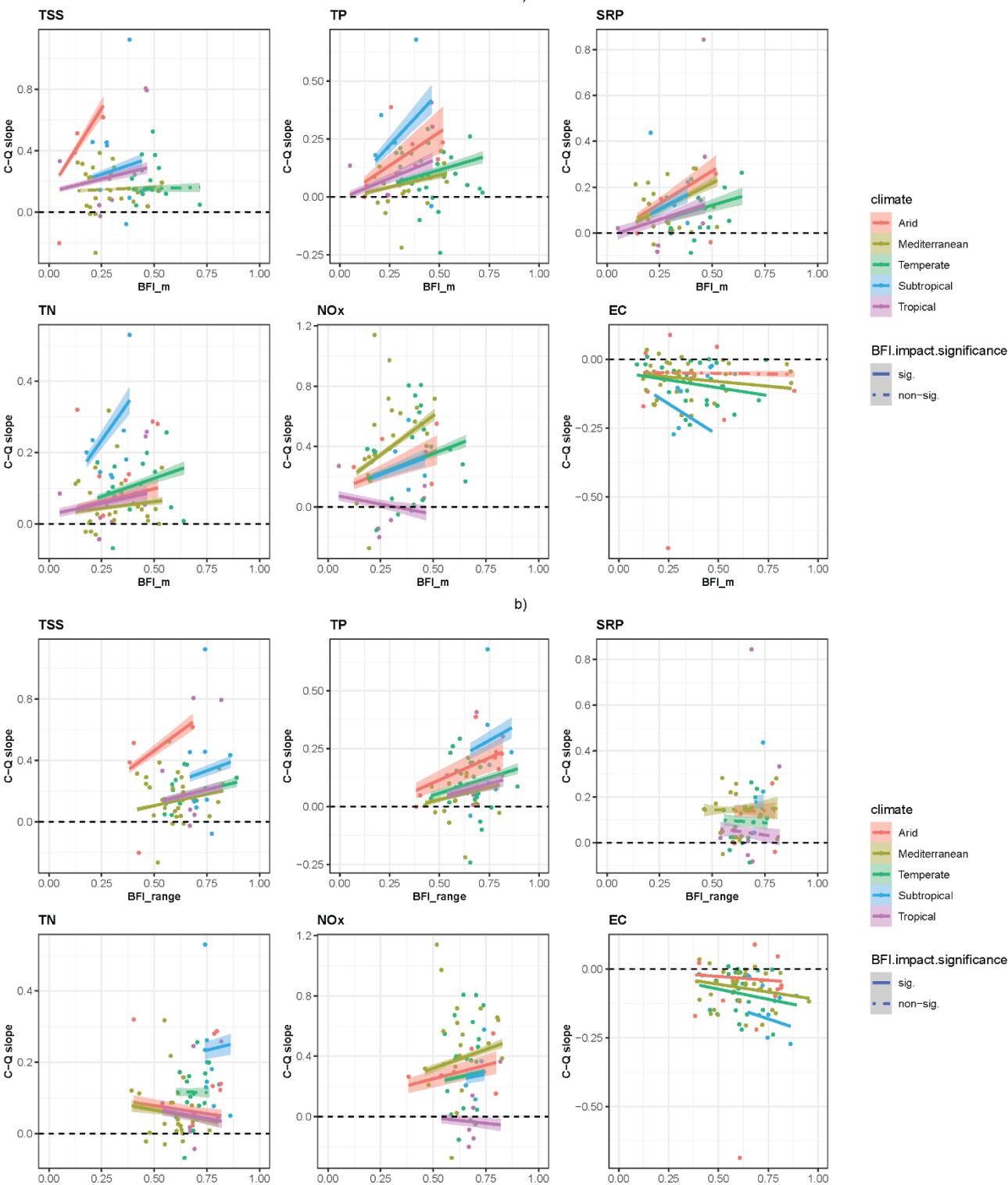

**Figure 6. a) Catchment C-Q slope vs. *BFI_m* and b) Catchment C-Q slope vs. *BFI_range*, coloured by climate zones. The lines represent the modelled C-Q slope~*BFI_m* or C-Q slope~*BFI_range* regression lines for individual climate zones. The bands represent the 95% credible interval (the range between 2.5th to 97.5th percentiles of Bayesian posterior distribution) of the modelled C-Q slopes. The dots represent the 'true' C-Q slopes estimated with C-Q observations at individual catchments. The black dashed lines mark a zero C-Q slope which differentiate mobilisation (C-Q slope>0) from dilution (C-Q slope<0).**

Figure 6 highlights the overall impacts of baseflow contribution on the C-Q slopes of individual water quality variables. For TSS, TP and $NO_x$, we generally see stronger mobilisation in catchments with higher values in both *BFI_m* and *BFI_range*. For TN, the mobilisation is stronger with higher *BFI_m*, while *BFI_range* generally has slightly negative effects. The detailed model results for each water quality parameter and potential interpretation are discussed in the following paragraphs; we do not further interpret the modelled results for SRP and EC due to limited ability to explain variation in their C-Q slopes (Table

1).

TSS and TP both show consistently strong positive effects of the baseflow contribution on the C-Q relationships, which both have increasing positive C-Q slopes with a higher value in either *BFI_m* or *BFI_range*, for most climate types. For TP, both *BFI_m* and *BFI_range* have significant positive impacts on the C-Q slopes for catchments across all climate zones, while for TSS, *BFI_range* always has a significant positive impact on the C-Q slopes and *BFI_m* has consistent impacts for arid,

subtropical and tropical catchments. These positive effects of both *BFI_m* and *BFI_range* on the largely positive C-Q slopes of TSS highlight that particulate transport may be enhanced by higher overall baseflow contributions, as well as by greater variability in baseflow conditions. TP is largely particulate-bound, as evidenced by SRP:TP ratios lower than 0.4 for most catchments across all climate zones (Figure S8). Therefore, the transport of TP is likely also enhanced by variations in baseflow conditions in the same manner as for TSS. Note that the overall positive effect of *BFI_m* in enhancing mobilisation is generally

significant despite that catchments with higher *BFI_m* generally have lower median concentrations of TSS and TP (Figure S6), suggesting relatively limited sources in catchments with high *BFI_m*.

The enhanced mobilisation of particulates (TSS and TP) with higher *BFI_m* is consistent with previous studies in European catchments, which also reported positive effects of BFI on the C-Q slopes of TSS (Moatar et al., 2017; Musolff et al., 2015). However, no physical interpretation of this result was discussed previously. Combining our modelled results of *BFI_m* together

with those of *BFI_range*, we are able to draw a plausible explanation that links particulate mobilisation with the two highly correlated baseflow metrics (Figure 3b). Specifically, catchments with lower *BFI_m* generally have narrower ranges of variation in instantaneous BFI (low *BFI_range*), which thus tend to always be dominated by surface flow regardless of dry or wet conditions (Catchment A, Figure 7). This can lead to a limited range of water sources with small variation of flow pathways that transport chemical species to rivers, resulting in a relatively stable export pattern across low and high flows in these

catchments. In contrast, catchments with higher *BFI_m* generally have higher variability in instantaneous BFI. This suggests higher variations in flow pathways, including surface flow dominance during wet periods and subsurface flow dominance during dry periods (Catchment B, Figure 7). Consequently, we contend that catchments with high *BFI_m* and *BFI_range* can

have a higher diversity of flow pathways and water sources, potentially leading to larger chemical gradients between groundwater-driven concentrations at low flow and surface flow-riven concentrations at high flow.

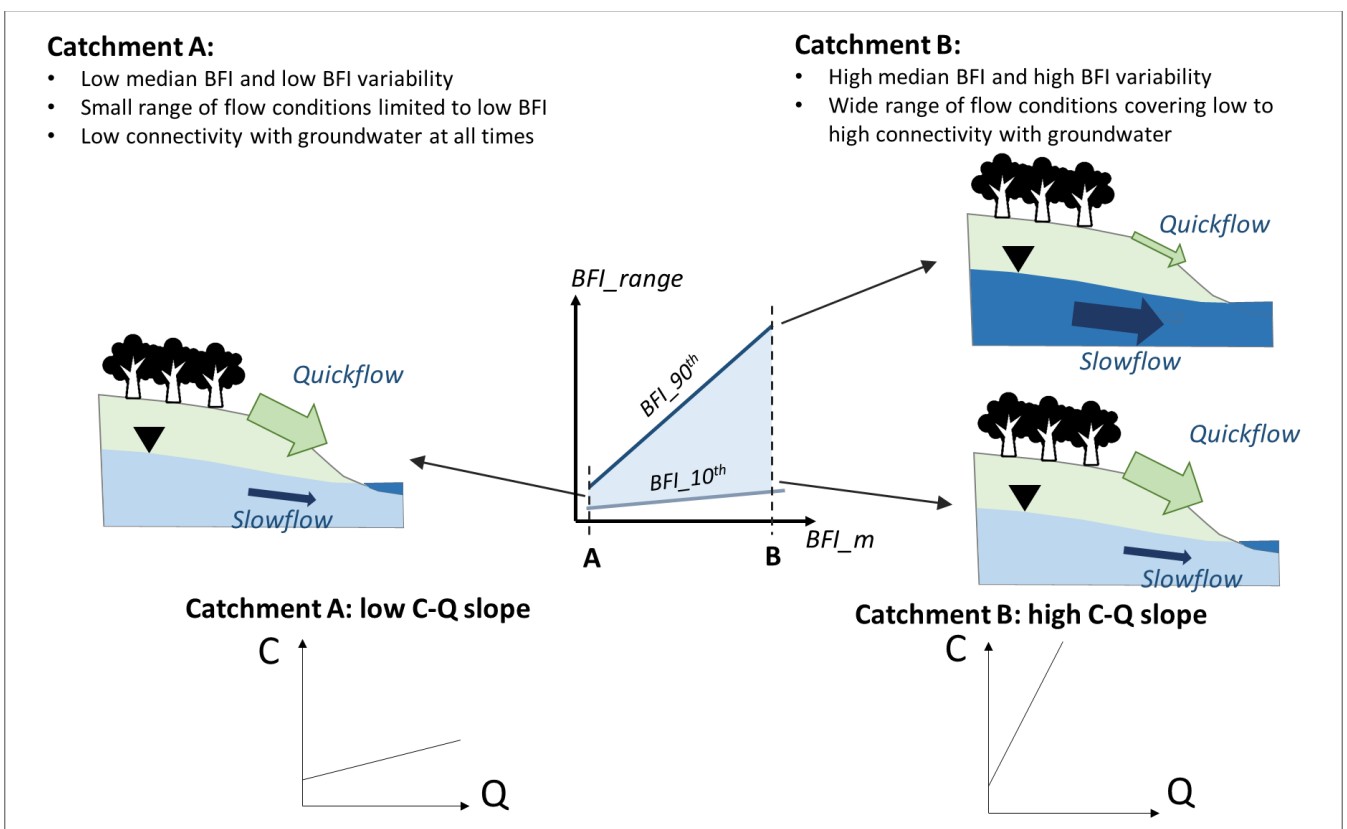

**Figure 7. Conceptual diagram of the expected hydrological conditions in catchments with low and high median and variability in baseflow contribution (*BFI_m* and *BFI_range*), as Catchments A and B, respectively. The contrasting hydrological conditions can help explain our modelled results of the impacts of baseflow contributions on C-Q slopes. Note that the C-Q intercepts in the plots are not indicative since we do not investigate the variation in C-Q intercepts in this study.**

For NO$_x$, the modelled effects are also largely consistent between models with *BFI_m* and *BFI_range* as the key predictor. The C-Q slopes increase (become more positive) with an increase in either *BFI_m* or *BFI_range*, for catchments within arid, Mediterranean, temperate, and subtropical climates, while they decrease for the tropics. This suggests enhanced mobilisation of NO$_x$ in most catchments when baseflow contributions and their temporal variations are higher. We note that tropical catchments generally have the lowest positive C-Q slopes, or even slightly negative slopes suggesting weak dilution export

patterns. Therefore, higher values in either *BFI_m* or *BFI_range* may actually enhance the dilution effects in tropical catchments, as opposed to mobilisation in other climate zones.

Soluble N and P concentrations in shallow groundwater are generally low in Australia (Cartwright, 2020). This contrasts with agricultural catchments in Europe and North America, where high nutrient levels are often observed in groundwater due to the

legacy of long-term agricultural practices (Van Meter et al., 2017; Stackpoole et al., 2019; Ehrhardt et al., 2018). Thus, it is also plausible that the enhanced mobilisation pattern seen in high *BFI_m* catchments is rather the effect of a high *BFI_range*. Catchments with a greater variability in baseflow contribution (Catchment B, Figure 7) are likely to have greater gradients of concentrations for soluble N, between low groundwater-fed concentrations at low flow and high concentrations from surface runoff and/or interflow at high flow (e.g., via leaching), resulting in a stronger mobilisation pattern as illustrated with a higher C-Q slope. Further, a typical temporal pattern of nitrate leaching in Australian catchments is the accumulation of N in soils during periods of low soil water drainage, followed by strong export during high drainage (Drewry et al., 2006), which is also more likely to occur in catchments with greater variation in baseflow contributions.

*BFI_m* and *BFI_range* show opposite impacts on the C-Q slopes of TN for all climate zones other than subtropical. A large proportion of TN in Australia is present in particulate forms (Figure S9) with most catchments having $NO_x$:TN ratios lower than 0.25, which contrasts with many other catchments in the United States and Europe (Ator et al., 2011; Durand et al., 2011). The particulate-dominated TN and the responses of C-Q slopes to *BFI_m* highlight that particulate N is largely mobilised across all climate zones, which are enhanced with higher contributions of baseflow. A stronger mobilisation is seen for high *BFI_m* across all climate zones, while a weaker mobilisation is observed for high *BFI_range* across all climates other than the subtropical catchments.

For the largely particulate-bound TN, one would expect the effects of *BFI_m* and *BFI_range* to be similar to TSS. This brings a question for the interpretation of the weaker mobilisation for a higher *BFI_range*. This unexpected result might suggest that the export patterns for particulate N are different to those for TSS and TP at various baseflow conditions. Therefore, further investigation is required on the impact of the variability in baseflow contribution on the export patterns of individual N constituents such as particulate N.

Besides the above-mentioned processes, another potential explanation for the modelled effects of *BFI_m* on C-Q relationships is related to flow seasonality, which needs to be further explored. High baseflow contribution is generally found in more perennial catchments in Australia (Kennard et al., 2010), which might be associated with more clearly defined seasonal patterns in transporting water quality variables. These conditions are likely leading to well-defined C-Q relationships (Minaudo et al. 2019) and could result in steeper slopes compared to other catchments. In contrast, catchments with lower baseflow contribution are more likely driven by intermittent flow while lacking clear seasonal patterns, which leads to more scattered C-Q relationships. In this case, the absolute values of C-Q slopes tend to be close to 0 and these catchments often fall in the category of chemostatic with unclear export regimes (e.g., Godsey et al., 2009). However, this hypothesis should be further tested beyond the current study, ideally with a subset of study catchments where high-frequency observations have been collected. Such future studies should also consider more broadly the temporal variations in flow regime and baseflow condition and their influences on C-Q relationships. For example, seasonality can play a big role in shaping the C-Q relationships for nutrients, as these relationships over time during the build-up of pollutant sources, and during the flushing of readily available

sources at the onset of high flow periods (Bende-Michl et al., 2013). Besides, anthropogenic disturbances and/or management actions in the catchment can cause changes in C-Q relationships over time (Zhang, 2018). Flow flashiness is also shown to influence the C-Q relationships, which differ across particulates and solutes, and across natural and highly regulated catchments (Moatar et al., 2020).

In summary, our results highlight the potential to improve understanding of transport processes via the relationships between water quality and baseflow contributions. Our model highlighted that the impacts of both *BFI_m* and *BFI_range* on catchment C-Q slopes are highly similar, which are both likely linked to the variability of baseflow contributions and thus the range of surface and subsurface sources and flow pathways. Therefore, to better understand the spatial variation of C-Q slopes, future research should seek alternative metrics to better capture the variability of baseflow conditions and the temporal dynamics of flow pathways within individual catchments.

## 4 Conclusions

In this study, a Bayesian hierarchical model was developed to understand the impacts of catchment baseflow contribution on C-Q slopes for six water quality parameters across Australia. Our model suggests significant influences of catchment baseflow contributions on C-Q slopes across most water quality variables and climate zones. Across the nation, the median and range of BFIs are also positively correlated for most studied catchments, while the C-Q slopes are largely positive for sediments (TSS) and both particulate and soluble N and P (NO$_x$, TN and TP). For TSS, NO$_x$ and TP we generally see stronger mobilisation in catchments with higher values in either the median or the range of catchment BFIs. The enhanced mobilisation in catchments with higher median BFI (and/or BFI range) is likely a result of more variable flow pathways over time, which introduces higher gradients of concentration between low and high flows. These low and high flows are likely dominated by different groundwater and surface water sources, each mobilising different pools of solutes and particulates. This result highlights the crucial role of flow pathways in determining catchment exports of water quality constituents, and the need for further studies to identify suitable hydrological metrics in differentiating flow pathways to improve the prediction and understanding of C-Q relationships. The results also suggest a priority for managing and monitoring stream P and N, which should focus on catchments with the greater fluctuations in baseflow contributions. To this end, it would be worth establishing explicit links between C-Q relationships and water age with high-frequency samples collected at select catchments (e.g., Cartwright, 2020).

This study used catchment-level metrics of baseflow contribution as the only predictor of C-Q slopes. The baseflow contribution alone can explain up to 22% variance in the C-Q slopes across the Australian continent. This highlights a substantial role in baseflow contribution in shaping the C-Q relationships, while also suggesting the need of further work to synthesise the impacts of baseflow contribution together with other spatial drivers (e.g., climate, land use, land cover and geology) to include their interactions and establishing their relative importance on influencing C-Q relationships. Further, this study used a linear model structure to synthesise large-scale patterns of the impacts of baseflow contribution on the C-Q

relationships across different climate zones. Although this model structure is limited and likely to be influenced by outliers, we believe it is suitable for the study purpose, as we are able to demonstrate the ability of the model to identify significant effects of catchment baseflow contribution on C-Q slopes, with statistically significant modelled effects for most climates and water quality parameters (Figures 5 and 6). Further studies can build on the learning from the current study to explore alternative model structures, to improve our ability to predict C-Q slopes within individual climate zones.

This study also highlights the effectiveness of Bayesian hierarchical models in interpreting water quality data across large spatial scales. Such a model is ideal to analyse water quality data over a large number of catchments, with high heterogeneity in temporal coverage and sampling frequency. This is particularly relevant for Australia, as water quality monitoring is often undertaken under different local/regional programs, and thus limited to certain timeframes and focusing on specific management interests.

### Data availability

Water quality and flow data used in this study are available upon request from seven Australian state agencies. These include: the Department of Land, Water and Planning (VIC DELWP, Victoria); WaterNSW (New South Wales); Department of Resources and Department of Environment and Science (QLD DNRME, Queensland); Department for Water and Environment (SA DEW, South Australia); Department of Water and Environmental Regulation (WA DER, Western Australia); Department of Primary Industries, Parks, Water and Environment (TAS DPIPWE, Tasmania) and Department of Environment, Parks and Water Security (NT DEPWS, Northern Territory). Sources of data are detailed in Section 2.1.1.

### Author contribution

All authors contributed to the design of the research. Danlu Guo carried out data collation, performed the simulations and prepared the manuscript with contributions from all co-authors. All authors contributed to the interpretation of the results and provided feedback.

### Competing interests

The authors declare that they have no conflict of interest.

### Acknowledgement

We would like to acknowledge that all water quality and flow data used in this study were accessed from the monitoring data collected by state agencies across Australia. In collecting these data, we were assisted by: Steve Tickell and Yuchun Chou

(Department of Environment, Parks and Water Security, Northern Territory), Erinna Colton and Ashley Webb (WaterNSW, New South Wales), Christine Webb (The Department of Water and Environmental Regulation, Western Australia), David Waters, Belinda Thomson, Ryan Turner, Rae Huggins (Department of Resources and Department of Environment and Science, Queensland), Bryce Graham (Department of Primary Industries, Parks, Water and Environment, Tasmania), Matt Gibbs (Department of for Water and Environment, South Australia), Paul Wilson (Department of Land, Water and Planning, Victoria). We also thank Professor Andrew Western, Ms Natalie Kho and Dr Rémi Dupas for their generous assistance in guiding this work and in data processing. Clément Duvert thanks Charles Darwin University for supporting a research visit to the University of Melbourne in March 2020. We also acknowledge the Traditional Owners who were the first custodians of Australian waterways.

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
