# Peer review of "Synthesizing the impacts of baseflow contribution on C-Q relationships across Australia using a Bayesian Hierarchical Model"

_Hydrology and Earth System Sciences, 2021_

## Author Comment (AC1)

**Responses to Reviewer #1 (hess-2021-353)**

Our responses are in blue and proposed revisions are underlined.

**Summary:**

1. Guo et al. investigate the variability in C-Q relationships in relation to the catchment hydrological conditions (more specifically the BFI) for several water quality parameters across several climate regions in Australia. The authors make use of an impressive data set from an arid region and apply a Bayesian Hierarchical Approach including the BFI, which allows understanding spatial patterns of export dynamics. This study can thus provide an important contribution for understanding solute transport beyond temperate regions. However, the manuscript still needs more clarity on the research questions and key messages and methodological improvements. I would suggest the manuscript for major revisions as substantial improvements are still necessary.

Thank you for acknowledging the value of the study, and your comprehensive suggestions for further improvement. We provide a point-to-point response to your comment below.

**General comments:**

2. One of my concerns is that given the fact that previous studies in the same region, using the same dataset and similar methods (as I read from the text), are not accessible or provided (under review or in preparation), it is not possible to judge the additional value of this study. The preceding studies (Lintern and Liu) are referenced both when defining the research goals and in the method sections. I definitely see the value of investigating C-Q relationships in various climate zones, but this was also done by these referenced studies. It is hard to judge the additional value without knowing what was shown already.

I believe this comment refers to L65-69 in the Introduction, which introduced our preceding studies, Lintern et al. (in review) and Liu et al. (in preparation), which both focused on how C-Q relationships vary across the Australian continent. Considering the paper timelines, we propose to keep only Lintern et al. (in review) here and add a summary of this study in the revised manuscript as:

- *"Lintern et al. (in review) focused on differences in water quality and the C-Q relationships across different climate zones. One remaining question that Lintern et al. (in review) highlighted is the high variation of the C-Q relationships within each climate zone, which required further explorations."*

We also propose to add the following discussion on new insights added by this current study to the end of the Introduction, as:

- *"We use a subset of the grand dataset that Lintern et al. (in review) used, which enabled us to focus on the representative observations for a wider range of flow conditions at individual catchments. As such, this study by analysing the impacts of baseflow contribution, adds new insights to explain the variations in C-Q relationships within individual climate zones."*

We also provide a copy of Lintern et al (in review) as a supplementary document for the reviewer's consideration. Note that Lintern et al (in review) is likely to be published before the publication of the current study, in which case we will include the full reference in the revised manuscript.

3. The motivation of investigating BFI impact on C-Q relationships was not convincing for me. It needs to be clear 1.) why we need to know that and 2.) what exactly we do not know yet. The first question is not satisfyingly presented: Why do you want to focus on BFI, why is it useful to

investigate this relationship? For the second: From my knowledge and in contrast of what you state (see also my comments below), the influence of BFI on the spatial variability of C-Q relationships has been discussed in several previous studies. However, I agree that studies have been biased towards temperate climates. I think the latter should be the main motivation, while generally the literature review on the control of BFI needs to be extended. It is not right, that is has not been investigated. There are studies using BFI as a descriptor for explaining the variability in export behaviour of different solutes, including several studies that you have cited in the introduction but considering other statements. For example, Minaudo et al. 2019 stated, "we found for NO3− that high BFI values, low W2, and low erosion differentiated C-Q dilution patterns from non-significant and mobilization types". But also Ebeling et al. 2021, Moatar et al. 2017, Musolff et al. 2015 have used the BFI to explain variability in C-Q relationships among catchments for several solutes. Moatar et al. 2020 also investigated the impact of discharge flashiness on C-Q slopes and subsequently load flashiness. As BFI and Q flashiness are closely linked, this needs to be mentioned in the introduction. These also need to be discussed in relation to your study in the discussion. Also see further comments below

This study is important because understanding the impact of baseflow contribution on CQ relationships can help explain both the spatial and temporal variation of CQ relationships, and to infer key transport pathways.

We agree with the reviewer that previous studies have investigated how variation in baseflow contribution affects CQ relationships across multiple catchments. The link between CQ patterns and interannual baseflow index (or similar metrics) were well studied (e.g., Moatar et al. 2017, 2020, Musolff et al. 2015, Ebeling et al. 2021), while Minaudo et al. (2019) explicitly explored the link between instantaneous BFI and CQ relationships. However, the existing studies on the impact of baseflow contribution are largely focused on the temperate climate. Therefore, we haven't yet well understood the effects of baseflow contribution across a wider range of climate zones, nor been able to quantify this impact.

The key motivation of this study is to understand and quantify how catchment-level metrics of baseflow contribution affect CQ relationships across catchments in a wide range of climate zones. Through synthesizing continental water quality data, we were able to gain large-scale understanding of how CQ changes in catchments with different hydrological regimes.

To clarify these, we propose to revise the Introduction to:

1) highlight the importance of the key issue that this study explores (impact of catchment baseflow contribution on the CQ relationship).
2) revise the literature review and add discussions on the state-of-art research on the impacts of baseflow contribution, including existing studies on the variation of CQ relationships across catchment as well as the role of flow flashiness.
3) improve clarification of the knowledge gap on the lack of understanding of how CQ relationships vary with baseflow contribution across a wide range of climate zones.

4.

a) Some of the methods seem inappropriate, especially as there is too few data for some climate-solute combinations to fit robust models/regressions and interpret them (further comments below).

b) Besides, I do not see the value of investigating the BFI impact within each climate zone individually, i.e. separating the climate zones and fitting different models, instead of investigating the BFI impact across the whole climate variability. I think it would be more

valuable to know what effect the BFI has across the whole climatic variability, i.e. the continuum of variations. The climate zones, could be represented by their characteristics such as precipitation amount, seasonality, aridity, temperature etc.. Even within the climate zones those variables vary and could potentially explain the deviations not explained by BFI.

a)

In this study, we used a single Bayesian Hierarchical Model (BHM) to fit all data points for each constituent simultaneously, across all climate zones. Although we used different parameters for individual climate zones, this model structure has a great advantage of 'borrowing power', meaning that information is shared across climate zones, and parameter fitting for one climate zone can utilize data from all other climate zones. As illustrated in Figures 5 and 6, this approach has been effective in estimating the impact of BFI_m (catchment median BFI) on CQ slopes for climate zones with limited data, while acknowledging the greater uncertainty. This model enables us to estimate the BFI impacts for climate zones with limited data by making use of data from other climate zones; in this way we made the best use of all data available. BHM is thus ideally suited to data-limited situations in dealing with spatial data, which was also illustrated in our previous works (Guo et al., 2018; Guo et al., 2020; Liu et al., 2021).

We plan to add further discussion to the Method section on why BHM is suited to the dataset analysed in this study. Specifically, in the introduction paragraph of Section 2.2 (which introduces the model), we will add discussions on the key advantages of BHM in effectively handling data-limited situations and spatio-temporal data with uneven coverage. We will also add references to our previous works which have illustrated these advantages.

References:

Guo, D., Lintern, A., Webb, J. A., Ryu, D., Bende-Michl, U., Liu, S. & Western, A. W. (2020). A data-based predictive model for spatiotemporal variability in stream water quality. Hydrology and Earth System Sciences, 24(2), pp. 827-847.doi:10.5194/hess-24-827-2020

Guo, D., Lintern, A., Webb, J. A., Ryu, D., Liu, S., Bende-Michl, U., . . . Western, A. W. (2019). Key Factors Affecting Temporal Variability in Stream Water Quality. Water Resources Research, 55(1), 112-129. doi:10.1029/2018wr023370

Liu, S., Ryu, D., Webb, J. A., Lintern, A., Guo, D., Waters, D., & Western, A. W. (2021). A Bayesian approach to understanding the key factors influencing temporal variability in stream water quality – a case study in the Great Barrier Reef catchments. Hydrol. Earth Syst. Sci., 25(5), 2663-2683. doi:10.5194/hess-25-2663-2021

b)

The reason to investigate BFI effects for individual climate zones separately is to see whether we could identify any statistically significant differences of the effects for different climate zones, and the results suggested that this is the case (from the CQ slopes in Figure 5 and the simulated CQ~BFI relationships in Figure 6). If there is no significant difference of the BFI effects between climate zones, the model is also capable to indicate this – as would be shown with similar, undistinguishable modelled effects of BFI for individual climate zones. Our current model structure enables us to test this hypothesis, since we are essentially developing different models for each climate zone and comparing them within one larger modelling framework. In contrast, a model with only a single parameter for the BFI effects across all climate zones cannot be used to test this hypothesis.

We further explored the performance of a model which only allows a single parameter for the BFI effects across all climate zones. As shown in Table R1, a model that uses a lumped effect of BFI_m across all climate zones can barely explain any variation in the CQ slopes. However, we see this as a useful result to include and to illustrate any value added by exploring climate-specific effects of BFI. Therefore, we propose to include in Section 3.3.1 a new comparison of performance between the lumped and climate-specific models. This new comparison will replace the existing results in Section 3.3.1 which have caused confusion (as highlighted in your Comments #24 and #26, see our further responses to those comments).

**Table R1. The performance in explaining variation in the CQ slopes (summarized by R2), for a model that only allows a single effect of BFI_m across all climate zones.**

| WQ variable | Baseline (lumped) |
|-------------|-------------------|
| EC | 0 |
| TP | 0 |
| FRP | 0.03 |
| TSS | 0 |
| NOx | 0.03 |
| TN | 0.02 |

We agree with you that variations in the CQ slopes can also be explored together with other climate drivers (e.g., temperature, rainfall, aridity), but a key challenge we see in other water quality modelling studies with these potential drivers is the high-cross correlation between these variables, thus we decided to use climate zones as an integrator of many hydroclimatic variables. The focus of this study is BFI and we hypothesise that BFI has a significant effect on CQ slopes – which is supported by the study results. To address your comment, we will add further justification to the use of climate zones in Section 2.2 (Methods) for exploring the BFI effects. We will also add discussions on the need for further studies on the impact of individual climate drivers on CQ slopes in the Conclusion.

5. The interpretation of BFI_m in terms of variability of flow paths is not convincing to me as you could easily and more directly use the range of BFI to determine the relationships between C-Q slope variability with BFI ranges. I think it would be good to look at this instead of speculating, as you have the data at hand and Figure 3b is not convincing enough for this interpretation, in my opinion. Instead of the range BFI_h-BFI_l you could also consider other metrics of variability.

To address this comment, we performed further analyses to explore the impact of catchment BFI range on the CQ slopes. Specifically, we explored a new model with the same structure as our original model based on BFI_m, which is now based on BFI_range (i.e. the difference between BFI_h and BFI_l). The modelled effects of BFI_range on CQ slopes (see Figure R1 below) are highly consistent with the modelled effects of BFI_m on CQ slopes (see Figure R2 below, with minor updates from the original Figure 6 after correcting some previous errors in NOx data). These new results provide more concrete evidence on our previous 'speculation' on how the BFI range in a catchment can impact its CQ slope. We will add the new modelled effects to the Results section and update our discussion accordingly.

We also further investigated on the non-linear relationship between BFI_m and the BFI_range beyond the current results in Figure 3b) in the manuscript. We found that the non-linearity is a result of BFI_l and BFI_h both increasing with BFI_m, but the increase in BFI_h slows down at high BFI_m (see Figure R3 below). This result further highlights the need to explicitly consider the impact of the BFI_range. Thus, in addition to the proposed addition of the BFI_range model, we also plan to replace the current

[Figure]

Figure 3b) in the manuscript with Figure R3, which justifies the need of a separate model with BFI_range as the predictor.

**Figure R1. Catchment C-Q slope vs. catchment BFI range (BFI_range, as the difference between BFI_h and BFI_l), coloured by climate zones. The lines represent the modelled C-Q slope~BFI_range regression lines for individual climate zones,**

[Figure]

**Figure R2. Catchment C-Q slope vs. catchment median BFI (BFI_m), coloured by climate zones. The lines represent the modelled C-Q slope~BFI_m regression lines for individual climate zones,**

[Figure]

**Figure R3. The low (10th percentile) and high (90th percentile) of instantaneous BFI (BFI_l and BFI_h), and the range of instantaneous BFI (BFI_h - BFI_l) versus BFI_m. The plot includes all 157 catchments across six water quality variables studied.**

6. Linked to the methodology, I have a concern about the conceptualisation in Figure 7 and main conclusions. I think some methodological approaches and results/evidence are not robust and clear enough to generalise the results in the given way.

This figure will be heavily revised to incorporate our new results with the model that focuses on the impact of BFI_range on catchment CQ slopes (as detailed in our response to your Comment #5). To further improve the clarity, we will remove the panels on specific constituents to simplify the plot, and also revise the organization of the figure to improve the connection of individual panels.

7. The discussion misses comparison to relevant previous studies. Previous studies investigating hydrological controls (such as the BFI, flashiness etc.) spatial variability of C-Q relationships. The discussions needs extension

To address this comment, we propose to add some in-depth discussion of the studies summarized in the Introduction, which looked at various hydrological controls of CQ relationships and how they vary across catchments. We will highlight the key findings of these studies, compare our results to the existing studies, and discuss the added value of our work.

Specific comments:
8. L17: Does the baseflow contribution in a catchment impact the concentration itself or the C-Q relationship? For me, it seems like spatial and temporal dimensions are mixed up here.

Here we aim to provide a general statement of the importance of understanding C-Q relationships. For generality and clarity, we will remove the 'spatial and temporal variation' and revise the sentence as:

- *'Understanding the concentration-discharge (C-Q) relationships informs solute and particulate export processes'*

9. L18: This is not true "these patterns have not yet been investigated across large spatial scales", e.g. Minaudo et al. 2019 (see also related comments)

We propose to revise this sentence as:

- *"Previous studies have shown that the extent to which baseflow contributes to streamflow can affect C-Q relationships in some catchments. However, existing understanding on the effects of baseflow contribution in shaping the CQ pattern is largely derived from studies in temperate areas, while we still lack quantitative understanding of these effects across a wide range of climates."*

10. L48: "variable, which" reference is unclear, you mean here the studies? Please revise

We propose to start a new sentence from "which", as:

- *"These studies highlighted a number of critical drivers for these spatial variations, such as land use, land management, lithology and topography (e.g., Ebeling et al., 2021; Minaudo et al., 2019)."*

11. L 57: This sentence needs revision. The concentration variability within one catchment regarding the contribution of baseflow or quickflow to the current discharge is represented by its C-Q relationship, i.e. the variability "within a particular catchment". However, the cited studies also investigate the differences in C-Q relationships among catchments. Therefore, the provided references do not fit to this sentence, in my opinion. This also leads to the next sentence being incorrect. It defines the research gap as the differences among the catchments regarding the hydrological average behaviour not being investigated and understood. E.g. Minaudo et al. 2019 (others, see main comment) considered BFI in the analysis of C-Q relationship variability among catchments.

We agree with the reviewer that previous studies have investigated how variation in baseflow contribution affects CQ relationships across multiple catchments. As in our response to your Comment #3, we acknowledge the key gap that this current study addresses is the lack of understanding of the effects of baseflow contribution on CQ slopes across a wider range of climate zones, as well as the lack of quantification of the impact of baseflow. The key motivation of this study is thus to understand and quantify how catchment-level metrics of baseflow contribution affects CQ relationships across catchments in a wide range of climate zones.

We propose to revise the Introduction (including this sentence) to:

1) highlight the importance of the key issue that this study explores (impact of catchment baseflow contribution on the CQ relationship).
2) revise the literature review and add discussions on the state-of-art research on the impacts of baseflow contribution, including existing studies on the variation of CQ relationships across catchment as well as the role of flow flashiness.
3) improve clarification of the knowledge gap on the lack of understanding of how CQ relationships vary with baseflow contribution across a wide range of climate zones.

12. L91: "nitrate-nitrite" I am not sure what you mean with that. Is it the sum of both nitrate and nitrite concentrations? It should be defined once in the manuscript

Thank you for highlighting this, we will add this in the revised manuscript as:

- *"... total suspended solids (TSS), total phosphorus (TP), soluble reactive phosphorus (SRP), total nitrogen (TN), the sum of nitrate and nitrite (NOx) and electrical conductivity (EC)"*

13. L105 "unaffected" this is a strong word, I suggest to say "more robust"

We will revise this phrase as:

- *"...this is to ensure that the C-Q relationships are more robust with the presence of outliers..."*

14. L106: plural "span"

Thank you. We will revise this phrase as:

- *"Having water quality time-series that span at least 3 years..."*

15. L111: "met the above criteria across all the six water quality variables" for me this sounds as if the stations needed to meet the criteria for all the six variables, which was not the case from what I read in the next sentence and the following. I suggest to revise this formulation

We will revise this sentence as:

- *"We performed the above catchment selection for each water quality variable and found a total of 157 sites (catchments)."*

16. L144: Does that mean you fit equation 1 only to baseflow discharge? How can this work, if C is a mix of baseflow and quickflow concentrations? This sentence in unclear, please revise.

No, Eqn 1 was fitted to all available pairs of C-Q data regardless of whether the flow is baseflow dominated or quickflow dominated. To clarify this, we propose to revise the sentence in L145 as:

- *"Our model is based on such a single C-Q relationship at individual catchments. However, our model enables the slope term ($\beta_s$) for different catchments to change with the overall catchment-level baseflow contribution. Such model conceptualization is based on previous literature on the effects of baseflow contribution on C-Q slopes within individual catchments (Gorski & Zimmer, 2021; Minaudo et al., 2019), while aiming to further explore the impact of BFI across multiple catchments."*

17. L146: "the C-Q slopes of all catchments are following a normal distribution with a 'grand mean'" This works if the represented catchments cover the range of variability well. This would not be true if catchment types are overrepresented, would it?

The use of this 'grand mean' for each water quality variable is based on a finding from our preceding study, that the C-Q slopes for each water quality variable do not vary substantially across climate zones (Lintern et al. in review). We clarify that we used the mean in our model to represent a baseline level of CQ slopes for each water quality variable; whereas the CQ slope of each catchment is the sum of the grand mean and the BFI effect (rather than sampled from a normal distribution), following Eqn 2. To clarify this, we propose to revise this sentence to a more accurate description of the model structure as:

- *"We assume that for each water quality variable, the C-Q slopes of all catchments have a 'grand mean', β0. Then the variation of C-Q slopes between catchments, away from β0, are explained by changes in catchment BFI."*

As a further comment to the representativeness of catchments that we analysed, we acknowledge the lack of data for some regions due to limited monitoring (i.e., regions of low management prioritization/interest for monitoring, and regions inaccessible for monitoring). Further, climate zones and catchments are unequally distributed across Australia; catchments are more clustered towards the coastline due to higher rainfall and topography, while the large arid region in the middle of the

continent does not have many flowing rivers. These mean that we expect the catchment representation to be biased towards regions that have more catchments and are monitored more frequently (e.g., in the coastal regions of Australia).

18. L146f: "Then the variation of C-Q slopes between catchments, away from beta_0, are explained by changes in catchment BFI. " This would only be true, if the BFI is the "only" controlling variable

We intend to use BFI as the only controlling variable because this study focuses on the impacts of BFI on the variation of CQ slopes across catchments. To clarify, we will add a clarification of this assumption after this sentence:

- *"This model conceptualization also assumes that the BFI is the only controlling variable and thus will inform how well BFI explains the spatial variation of C-Q slopes."*

19. Fig2: I suggest to add axis labels and ticks for panel a and b

We will revise the figure as suggested.

20. L185: "together with flow" I do not understand this. If c=f(Q) in the C-Q relationship this is already included.

We will delete this phrase. This entire paragraph 'Model performance' will be replaced with the new model performance comparison to be presented in Section 3.3.1 (comparison of performance between the lumped and climate-specific models, as detailed in our response to your Comment #4a).

21. L214 "surface flow" is imprecise. There is not just baseflow and surface flow

We will replace this phrase as 'quickflow' for consistency with the rest of the paper.

22. L217: "In contrast, a catchment with a high BFI_m generally has a large range in instantaneous BFIs" I see several high BFI_m with not very high ranges in BFI. Fig3b rather looks like a bell shape with highest ranges for medium values, not like a linear relationship.

We agree with your observation. This nonlinearity highlights the need to explicitly consider the impact of the range of BFI. This is further justified with our additional investigated on the non-linear relationship between BFI_m and the BFI_range as below in Figure R3. We found that the non-linearity is a result of BFI_l and BFI_h both increasing with BFI_m, but the increase in BFI_h slows down at high BFI_m. As detailed in our response to your Comment #5, we plan to update the current Figure 3b) as Figure R3, as well as adding an alternative model structure with the range of BFI as the key predictor, so that we can explicitly consider the impact of the range of BFI on CQ slopes.

[Figure]

**Figure R3. The low (10th percentile) and high (90th percentile) of instantaneous BFI (BFI_l and BFI_h), and the range of instantaneous BFI (BFI_h - BFI_l) versus BFI_m. The plot includes all 157 catchments across six water quality variables studied.**

23. Fig 3: Boxplots can create confounding impressions if the sample size is very different, which I see from Figure 1. I would suggest adapting the boxplot widths according to the sample size and/or writing the sample size number to the plot for each climate zone (probably the numbers in the x axis labels?). For the right panel, I would suggest to colour the dots according to their climate zone as in Figure 1, possibly also for the left panel with different hue or saturation values to distinguish the different BFI quantiles.

Thank you for the suggestions. We already have the sample size for each climate zone labelled on the x-axis, we will add a sentence to the figure captions to clarify this.

We will also use different colours for individual climate zones in both the left and right plots as suggested.

24. L255: "BFI-based model has only marginally lower performance … BFI-based model, while having the capacity to predict C-Q slope across space, can predict water quality almost as well as using the observed C-Q slope." I do not understand this comparison of individual C-Q slope with a model explaining the variability in C-Q slopes. It sounds like you were expecting worse performance, while actually a more complex model should improve performance to be a valid approach.

As detailed in our response to your Comment #4b), we propose to remove the current results in Section 3.3.1 and instead present new results that compare the performance of models with i) climate-specific impact of BFI; and ii) a single impact of BFI across all climate zones. This update will hopefully clarify our approach and avoid any confusion on our communication of model performance.

25. L269: "confidence intervals"? I do not know credible intervals

"Credible interval" is the term used to describe uncertainty in the posterior samples (including simulations and parameter values) from Bayesian modelling (Gelman et al., 2013). We propose to add a sentence to clarify this as:

- *"The significance of the modelled effect of catchment median BFI is assessed by the 95th credible intervals (the range between the 2.5th and 97.5th percentiles of the posterior parameter values, which measured the uncertainty in Bayesian posterior distribution, as Gelman et al., 2013). The effect of catchment median BFI on C-Q slope is almost always significant, with the 95th credible intervals not crossing over 0 for most combinations of water quality variables and climate zones."*

Reference: Gelman, A., Carlin, J. B., Stern, H. S., Dunson, D. B., Vehtari, A., & Rubin, D. B. (2013). Bayesian Data Analysis, Third Edition: Taylor & Francis.

26. Table 1: Why do you think the NSE of your BFI-base model are lower than the baseline model? Does it actually make sense to fit a more complex model in this case?

As detailed in our response to your Comment #4b), the current results in Section 3.3.1 will be removed and replaced with new results that compare the performance of models with i) climate-specific impact of BFI; and ii) a single impact of BFI across all climate zones. We believe this update will clarify our approach and avoid any confusion on our communication of model performance.

27. Section 3.2 I do not like this whole section, I think the statements derived from selected examples are not representative for the catchments distribution of TSS export patterns within the corresponding climate zones (Fig6)

We intend to present these cases only as examples to show the potential impacts of BFI. We expect that showing the actual data of C, Q and BFI can better help the readers to understand the key model components and relationships we aim to represent. We plan to add a final sentence to this section to highlight the limitation of the examples and need to explore a wider range of catchments to generalize the pattern:

- *"However, the impacts of BFI seen in these examples cannot meaningfully represent large-scale patterns over a wide range of catchment conditions. Section 3.3 presents our modelled results in which these large-scale patterns are synthesized."*

28. Figure 5: What is the modelled effect? Is it dBFI_climate from equation 3? What is the NSE above each subplot describing?

Yes, the modelled effect is $\delta BFI_{climate}$ in Eqn 3. This is already described in L269 before the Figure as: "Figure 5 presents the median and the 95th credible intervals of these modelled impacts for each water quality variable, derived from the Bayesian posterior estimates of $\delta BFI_{climate}$ (Eqn. 3)."

The NSE above each panel summarizes the model performance to predict the concentration of individual water quality variables. We will revise the figure captions to clarify these points as below. Note that the term $\delta BFI_{climate}$ will be revised as $\delta_{BFI\_climate}$ throughout the manuscript to avoid the confusion highlighted in Comment #10 of Reviewer 2.

- *"Figure 5. Modelled effects of BFI_m on C-Q slope for each climate zone, i.e., $\delta_{BFI\_climate}$. The bars show the 95% credible interval (2.5th to 97.5th percentile) of modelled effect for each climate zone for each water quality variable, and the dots indicate the corresponding median levels. The colours indicate whether an effect is significantly positive, significantly negative, or non-significant; a positive effect means that C-Q slope increases with higher catchment BFI,*

*and vice versa. Black dashed lines show zero-effect i.e. no effect at all. The NSE above each panel summarizes the model performance to predict the concentration of individual water quality variables."*

29. L278-285: I do not agree with this approach and subsequent observation. Fitting linear relationships is not appropriate for the given observations and "consistent diverging" behaviour goes beyond what can be interpreted here. Especially for the last two sentences: When looking at the overall point clouds, there is not clearly increasing variance (diverging behaviour) with higher BFI_m. In my opinion, weak relationships are overinterpreted here. These are also transferred to conclusions.

We'd like to clarify that L278-285 do not intend to make inference on the BFI effects here; instead, we clarify that the diverging patterns are a result of the model structure. To clarify, we propose to add a further sentence after L285 as:

- *"Since these diverging patterns are a result of the model structure, we do not further interpret any physical representations on the impacts of BFI."*

We have not included any summary of these results in the Conclusion section.

We understand the concern about the limitation of the linear model used, but we consider this model structure to be appropriate for our study purpose, which is to synthesize large-scale patterns of BFI impacts across different climate zones. The model structure is suitable for our purpose, with demonstrated ability to identify significance effects of catchment baseflow contribution on CQ slope (i.e., these modelled effects are mostly constrained to a very limited range as seen in Figure 5 in the manuscript). We do acknowledge the limitation of this structure and the potential issue of outliers, and the need of further studies to explore model improvement possibly via more complex model structures - we proposed to add further discussions on these points in the revised Discussion section.

30. L283: 'grand mean' I know you have used this term before, but I think it is not well chosen, as it does not tell that it is about the "solute-specific base C-Q slope". Consider changing the term

Thank you for the suggestion, we will revise this term throughout the manuscript.

31. Figure 6: In my opinion, it is not justifiable to fit linear regressions to all combinations of solutes and climate zones, because several combinations have 1) too little sample sizes, 2) clearly non-linear relationships, and 3) in some cases plus influenced by outliers, e.g. the Tropical NOx fit. The legend titles should be improved.

Firstly, small sample size for some climate zones is not a real limitation of the Bayesian Hierarchical Model due to its ability to use information across data groups, as explained with more details in our response to your Comment #4a). We also propose to add more details on the advantage of the Bayesian Hierarchical Model in working with small sample size in the Method section. Specifically, in the introduction paragraph of Section 2.2 (which introduces the model), we will add a brief justification for applying BHM. We will discuss the key advantages of BHM in effectively handling data-limited situations and spatio-temporal data with uneven coverage, which are illustrated in our previous studies (Guo et al., 2018; Guo et al., 2020; Liu et al., 2021).

Secondly, we consider a linear model suitable for the purpose of this study, which was to synthesize large scale patterns with a simple model structure applied to the whole country. The model structure is suitable for our purpose, with demonstrated ability to identify significance effects of catchment baseflow contribution on CQ slope (i.e., these modelled effects are mostly constrained to a very limited range as seen in Figure 5). We acknowledge the limitation of this structure and the potential

issue of outlier, and the need of further studies to explore model improvement possibly via more complex model structures. We will add discussions on this limitation and future studies to the revised Discussion section.

We will also add the legend title in the revised figure.

32. How does the modelled effect from Figure 5 relate to the slope of the linear regressions in Figure 6? This seems somewhat redundant to me.

The modelled effects in Figure 5 are the same as the slopes of the linear regression in Figure 6. Although the two figures overlap on the impacts of BFI, they focus on different aspects. Figure 5 focuses on the direction and significance of the BFI impacts themselves, while Figure 6 focuses on illustrating these effects with CQ slopes and BFI. Further, Figure 6 is a more intuitive way to show the impacts on BFI. Therefore, we believe it's necessary to keep both figures.

33. Figure 7: Firstly: The generalisation shown is questionable (see main and other comments). E.g. Figure 3b shows that highest BFI ranges are for medium BFI_m, suggesting that high BFI might also have more stable flow conditions with generally higher groundwater contributions. Secondly: This Figure takes a lot of time to understand and could benefit from some reworking including/according to other adaptations. The Figure text is unclear without reading the main text, as well as the meaning of a1, a2, b1, b2 only from the second reading. I suggest selecting other identifiers. The spatial organisation, e.g. link between the upper and the bottom panel and left, middle and right column, is not visually clear.

This figure will be heavily revised to incorporate our findings with the additional BFI_range model (as detailed in response to your Comment #5). For further clarity, we will remove the panels on specific constituents to simplify the plot, and also revise the organization of the figure to improve the connection of individual panels.

34. L332-334: This statement could benefit from checking also mean concentrations: are activated sources low or high in more quickflow dominated catchments?

We have checked this and included the results in Figure S11 in the original manuscript. In general, we see high median concentration at the more quickflow dominated catchments (note: since we work in the log-10 space for both concentration and flow in deriving the CQ slopes, the median concentration is a more useful indicator than the mean). To address this comment, we propose to revise this description as below to include the pattern on median concentration:
- *"catchments with lower BFI_m generally have narrower ranges of instantaneous BFI, which thus tend to always have low baseflow contributions (i.e. surface flow dominated) regardless of dry or wet conditions (Catchment a, Figure 7). This can lead to a limited range of water sources, with small variation of flow pathways to transport water chemistry to rivers, resulting in a relatively stable export pattern across low and high flows at these catchments. Note that these catchments with lower BFI_m tend to have higher median concentration (Figure S11), suggesting relatively high activated sources from these limited range of flow pathways'*

35. L398-391: I cannot follow this point unfortunately due to missing information. Moreover, the characteristics land use, geology and climate are all integrated in the base flow index, which is a resulting hydrologic characteristic. This point definitely also needs a discussion part, including further literature and potential controls.

We propose to delete this sentence as the referenced study (Liu et al., in preparation) is still ongoing. Instead, we will highlight the need of further investigating other potential drivers for spatial variation of the CQ relationships, such as land use, land cover, geology and climate.

36. L392: Why is the Bayesian hierarchical model more effective than multiple linear regressions with BFI or other multivariate models? For me, Figure 5 (outcome of the Bayesian model) and Figure 6 (not a direct model output) were somewhat redundant. This is not covered in enough in the discussion section.

Bayesian Hierarchical Model (BHM) and multivariate regressions are not exclusive to each other i.e. BHM is also capable of including multiple predictors. We are not intending to compare BHM with any other model with this statement, instead we highlight the efficiency of BHM in making inferences and synthesizing large-scale patterns with limited and heterogeneous data availability across space. We will revise this paragraph to better clarify these points.

Figures 5 and 6 are intended to show different aspects of the results although they have overlapped contents, we explained this in detail in our response to your Comment #32.

37. SUPPLEMENTS: Figure S3: I do not understand why the BFI_m should be shown per solute, if the BFI depends on discharge and not on concentration.

BFI depends on discharge only, but the catchments analysed for individual water quality variables differ because the catchment selection criteria (Section 2.1.1) considered water quality data availability. This figure shows each water quality variable in as a subplot to incorporate the difference in catchments analysed.

38. REFERENCES: Ehrhardt et al. is already finally published, please change reference to: Ehrhardt, S., Kumar, R., Fleckenstein, J. H., Attinger, S., & Musolff, A. (2019). Trajectories of nitrate input and output in three nested catchments along a land use gradient. Earth Syst. Sci., 23(9), 3503-3524. https://www.hydrol-earth-syst-sci.net/23/3503/2019/

Thank you, we will update this reference in the revised manuscript.

---

## Author Comment (AC3)

**Responses to Reviewer #2 (hess-2021-353)**
Our responses are in blue and proposed revisions are underlined.

This manuscript presents a synthesis of baseflow effects on C-Q relationships in watersheds across Australia. The authors have leveraged a Bayesian Hierarchical Model in this research. Overall, I think the research is solid, the manuscript is well written, and it can become an important contribution to the literature on riverine C-Q relationships. I provide below some comments to the author, which I hope can help improve the manuscript.

We thank you for providing valuable feedback on the study for further improvement. We provide a point-to-point response to your comment below.

1. The use of Bayesian Hierarchical Model should be more fully justified. I am aware of the research the team has done in the past few years involving Bayesian approaches, but why is it used in this work on C-Q relationships. Please provide your reasoning in the Introduction, probably the last paragraph.

Thank you for raising this point. We propose to improve justification of the Bayesian Hierarchical Models (BHM) as follows:

1) In the end of the Introduction, we will add a brief discussion on the uneven distribution of the available data across sites to highlight the value of applying BHM in this analysis.
2) In the introduction paragraph of Section 2.2 (which introduces the model), we will add a brief justification for applying BHM. Specifically, we will discuss the key advantages of BHM in effectively handling data-limited situations and spatio-temporal data with uneven coverage. We will also add references to our previous works (as below) which have illustrated these advantages.

References:

Guo, D., Lintern, A., Webb, J. A., Ryu, D., Bende-Michl, U., Liu, S. & Western, A. W. (2020). A data-based predictive model for spatiotemporal variability in stream water quality. Hydrology and Earth System Sciences, 24(2), pp. 827-847.doi:10.5194/hess-24-827-2020

Guo, D., Lintern, A., Webb, J. A., Ryu, D., Liu, S., Bende-Michl, U., . . . Western, A. W. (2019). Key Factors Affecting Temporal Variability in Stream Water Quality. Water Resources Research, 55(1), 112-129. doi:10.1029/2018wr023370

Liu, S., Ryu, D., Webb, J. A., Lintern, A., Guo, D., Waters, D., & Western, A. W. (2021). A Bayesian approach to understanding the key factors influencing temporal variability in stream water quality – a case study in the Great Barrier Reef catchments. Hydrol. Earth Syst. Sci., 25(5), 2663-2683. doi:10.5194/hess-25-2663-2021

2. The authors reported that the Bayesian Hierarchical Model can explain over half of the observed variability in concentration of TSS, EC and P species across all catchments (93% for EC, 63% for TP, 63% for SRP, and 60% for TSS). I feel the intention has switched here from understanding C-Q relationship to predicting water-quality concentrations, which seems to be a distraction to me. Moreover, what's the benefit of adopting the Bayesian Hierarchical Model, given that many statistical models (e.g., WRTDS) have been developed and can probably provide more accurate estimates?

Thank you. We propose to move the focus away from predicting water quality concentration, which can help better focus on our objective to understand the impact of BFI on CQ slopes. To achieve this, we will remove the current results in Section 3.3.1 and instead include new results to compare the performance of models with i) climate-specific impact of BFI; and ii) a single impact of BFI across all climate zones. This update will help to remove the previous confusion on our communication of the model performance, while also illustrating the value of considering climate-specific impacts of BFI in our study.

In addition, we acknowledge that our Bayesian Hierarchical Model (BHM) predicts across different catchments and is thus different to the catchment-specific WRTDS model. BHM is fitted to all catchments simultaneously and thus has a huge advantage of 'borrowing power' – to inform parameter estimation for one group of data by information from other groups (with 'group' being climate zone in our model). The hierarchical model architecture is ideally suited to grouped datasets, which enables data within the same group to share common features. The authors' previous studies illustrated the effectiveness of this model for simulating water quality temporal variability across multiple catchments (Liu et al., 2021; Guo et al., 2020; Guo et al., 2019). Therefore, BHM is useful to conceptualize the nation-wide C-Q dataset across multiple catchments and climate zones. We will further strengthen these to justify the use of BHM in the Introduction and Method sections (as detailed in our response to your Comment #1).

3. Figure 3b: It is not a strictly positive relationship for the entire range of BFI_m. The variability continues to increase with BFI_m up to ~ 0.5 and then starts to decrease with BFI_m. The latter part of the curve seems largely ignored in the manuscript, including Discussion. The same observation holds true for the individual constituents (Figure S5).

We agree with your observation. Our further investigation suggests that the non-linear relationship between BFI_m and the range of BFI is that both BFI_l and BFI_h increase with higher BFI_m, but the difference between BFI_l and BFI_h decreases at higher BFI_m > 0.6, which only occurs at a few catchments (see Figure R3 below). This highlights the need to explicitly consider the impact of the range of BFI. We plan to explore this by including an alternative model structure in the manuscript, where the range of BFI (BFI_range) is used as the key predictor. We also propose to replace the current Figure 3b) in the manuscript with Figure R3, which better explains the nonlinear relationship between BFI_m and BFI_range.

[Figure]

**Figure R3. The low (10th percentile) and high (90th percentile) of instantaneous BFI (BFI_l and BFI_h), and the range of instantaneous BFI (BFI_h - BFI_l) versus BFI_m. The plot includes all 157 catchments across six water quality variables studied.**

4.   It would be interesting to investigate the effects of seasons and antecedent discharge conditions (wet vs. dry), both of which may change the response of C-Q slope to the BFI_m metric. There may be strong contrast among, for example, growing vs. non-growing seasons. Toward the end of manuscript, the authors have briefly pointed out the possibility of season effects. I think it is probably beyond your scope to look into these effects in this paper, but I encourage the authors to provide a brief discussion to point out that the response of C-Q slope to the BFI_m metric can vary among different seasons, among different antecedent discharge conditions, and even among different periods. In the latter regard, it is reported that anthropogenic disturbances and/or management actions occurred in the catchment can cause the C-Q relationship to change. For example, Zhang (2018) provides an investigation of C-Q relationship for different river flows and years: https://doi.org/10.1016/j.scitotenv.2017.09.221.

Thank you for the excellent suggestion. We will add further discussions on the seasonal changes of BFI with respect to existing studies (including the suggested reference), and integrate these with our results to highlight further study directions.

5.   The term BFI_m (median BFI) is not self-evident in the Abstract. Given the importance of this metric, I encourage the authors to define it more clearly in the Abstract.

We will revise the abstract using 'catchment median BFI' throughout, instead of 'BFI_m'.

6.
   a)   The authors have used BFI_l and BFI_h to represent the variability of BFI, which makes sense to me. I may have used 2.5% and 97.5% instead but 10% and 90% are fine.
   b)   By the way, have you considered using standard deviation to capture the spread, which may help shorten the manuscript in terms of text and figures presented? I think an argument can

be added to the end of Section 2.1, which favors the use of BFI_l and BFI_h, that these are percentile based and hence are more robust to outliers.

a) Using 2.5% and 97.5% quantiles have potential risk to capture some outliers considering our catchment selection criteria (Section 2.1). Specifically, catchments with a minimum of 300 CQ-pairs can be included in our analysis, which means that the BFI_l and BFI_h are each if 2.5% and 97.5% quantiles are used to calculated BFI_l and BFI_h, each index can be calculated with <10 data points at the most data-scarce catchments. Therefore, we intend to keep the original decision of having BFI_l and BFI_h as the 10% and 90% quantiles of BFI for each catchment.

b) We thank you for suggesting an additional justification for the use of BFI_l and BFI_h and we will add this to the revised manuscript. A future relevant change we propose is to add a new model with the same structure as our original model based on BFI_m, which is based on BFI_range (the difference between BFI_h and BFI_l). The modelled effects of BFI_range of CQ slopes (see Figure R1 below) are highly consistent with the modelled effect of BFI_m (see Figure R2 below, with minor updates from the original Figure 6 after correcting some NOx data errors). These new results provide more concrete evidence on our previous 'speculation' on how the BFI range in a catchment can impact its CQ slope. We will add the new modelled effects to the Results section and update our discussion accordingly.

[Figure]

**Figure R1. Catchment C-Q slope vs. catchment BFI range (BFI_range, as the difference between BFI_h and BFI_l), coloured by climate zones. The lines represent the modelled C-Q slope~BFI_range regression lines for individual climate zones,**

[Figure]

**Figure R2. Catchment C-Q slope vs. catchment median BFI (BFI_m), coloured by climate zones. The lines represent the modelled C-Q slope~BFI_m regression lines for individual climate zones,**

7. For days with multiple samples, is it necessary to pre-calculate the average concentration? Why not keeping all the samples in the analysis? In addition, it may be helpful to provide a table that quantifies the fraction of such days in the record.

Streamflow records are generally in daily timestep (instead of sub-daily) across the nation, which limits us to use all high-frequency water quality samples to perform CQ analysis. To clarify this, we propose to add a sentence to justify this averaging process in the current Line 90 as:

- *"A daily average is taken if more than one water quality sample was collected for any day at any site. This is because streamflow records in Australia are largely in daily timestep, which limits our ability to analyse all high-frequency water quality samples."*

We will add a table to the supplementary information to specify the fraction of days with multiple samples – this generally occurs rarely except for some EC sites which collect high-frequency samples.

8. BFI calculation: I am curious about the use of 0.98 for alpha in the baseflow filter. Did Ladson et al. (2013) recommend this value? What is the rationale?

Yes, this was a recommendation in Ladson et al. (2013). The study stated: "recent comparisons of modelled and measured baseflow values in the Murray Darling Basin suggest a value of 0.98 produces more reasonable results.". Murray Darling Basin is a region which a large proportion of our study catchments are located (Figure 1), and we already clarified this with L130:

- *"The daily BFIs were estimated using a Lynne-Hollick baseflow filter with Alpha = 0.98 … as recommended for Murray-Darling Basin in the south-eastern Australia (Ladson et al., 130 2013), within which a large number of the study catchments are located*."

9. Figure 2: Please add numbers and units (even if hypothetical) on the y-axis for panels a and b.

We will revise the figure as suggested.

10. Equations 2-3: Consider changing $\delta BFIclimate$ to $\delta BFI\_climate$. (Move "BFI" to the subscript.) At first glance, I thought this is the product of two variables ($\delta$ and BFIclimate).

We will revise this term as suggested ($\delta_{\mathrm{BFI\_climate}}$) throughout the paper.

11. Section 3.3.1, including Table 1: I would like to refer back to my comment above. The NSE values do not seem to be comparable to more established approaches such as WRTDS. What is the value of showing these results? Should the baseline model or the BFI-based model be used for predicting concentrations? Why not those other established approaches?

As detailed in our response to your Comment #2, we propose to remove the current results in Section 3.3.1 and instead include new results to compare the performance of models with i) climate-specific impact of BFI; and ii) a single impact of BFI across all climate zones. We believe that this update will help to remove the previous confusion on our communication of the model performance.

Furthermore, since the model is fitted to all data across multiple catchments, this NSE value essentially describes how well the model explained the total spatio-temporal variability in water quality within the national dataset. Predicting spatio-temporal variability is a much more difficult task than predicting the temporal variability alone – as achieved by WRTDS for individual catchments (Zhang et al., 2021; Sprague et al., 2019). Thus we believe that the NSE values from BHM is not comparable with WRTDS. However, any related confusion should be removed with the proposed revision of Section 3.3.1.

References:

Zhang, Q., Webber, J. S., Moyer, D. L., & Chanat, J. G. (2021). An approach for decomposing river water-quality trends into different flow classes. Science of The Total Environment, 755, 143562. doi:https://doi.org/10.1016/j.scitotenv.2020.143562

Sprague, L. A., Mitchell, R. M., Pollard, A. I., & Falcone, J. A. (2019). Assessing water-quality changes in US rivers at multiple geographic scales using results from probabilistic and targeted monitoring. Environmental Monitoring and Assessment, 191(6), 348. doi:10.1007/s10661-019-7481-5

12. Section 3.3.2: According to published literature on many catchments around the world, SRP is a minor component of TP, whereas NOx is a major component of TN. It is quite interesting that in these Australian catchments, NOx/TN is quite small. This presents a strong contrast to many regions and may be discussed with a couple of sentences.

Thank you for highlighting this point, we will add some discussion on the contrasting NOx:TN pattern with other regions.

---

## Author Response (AR1)

**Response to Review Comments (hess-2021-353)**

**Response to the Editor**

Our responses are in blue and revisions are underlined.

1. I would like to commend the authors for their efforts to address the valuable comments that were received form the two reviewers. Given the extent of the revisions requested, I would appreciate if the authors would submit the revised paper that incorporates the changes they promised.

We have substantially revised the manuscript as promised. Please see below for our point-to-point response to the comments from both reviewers, along with our corresponding revisions.

2. I would also recommend that the STAN model be included in the supplementary material.

We have added the stan model codes to Figures S10-11 in the Supplementary Material.

3. My final recommendation is for all the uncertainty bounds in the figures be defined. I also have concerns about the weak slopes that appear to be influenced by a few high leverage points.

We have revised the figure captions to clarify the uncertainty in all modelled results, as:

- Figure 5. Modelled effects of BFI\_m and BFI\_range on catchment C-Q slopes for each climate zone ( $\delta_{BFI_cclimate}$ ) for each water quality parameter. The bars show the 95% credible intervals (the range between 2.5th to 97.5th percentiles of Bayesian posterior distribution) of the modelled effects, and the dots indicate the corresponding median levels. The colours indicate whether an effect is significantly positive (red), significantly negative (blue), or non-significant (grey); a positive effect means that the C-Q slope increases with a higher catchment BFI\_m or BFI\_range, and vice versa. Black dashed lines show the zero-effect i.e., no effect at all. The plot includes results from models with each of BFI\_m and BFI\_range as the key predictor, which are differentiated by marker shapes.
- Figure 6. a) Catchment C-Q slope vs. BFI\_m and b) Catchment C-Q slope vs. BFI\_range, coloured by climate zones. The lines represent the modelled C-Q slope~BFI\_m or C-Q slope~BFI\_range regression lines for individual climate zones. The bands represent the 95% credible interval (the range between 2.5th to 97.5th percentiles of Bayesian posterior distribution) of the modelled C-Q slopes. The dots represent the 'true' C-Q slopes estimated with C-Q observations at individual catchments. The black dashed lines mark a zero C-Q slope which differentiate mobilisation (C-Q slope>0) from dilution (C-Q slope<0).</li>

We also added clarification on model uncertainty in text and discussed its implication in Section 3.3 as:

- L327: "Figure 5 presents the median and the 95% credible intervals of these modelled effects for each water quality variable. The 95% credible interval is the range between the 2.5th and 97.5th percentiles of the posterior distribution of the parameter values, which was derived from the Bayesian posterior estimates of  $\delta_{BFI\ climate}$  (Eqn. 3) to quantify the uncertainty in the modelled effects (Gelman et al., 2013). The effects of BFI m and BFI range on the C-Q slopes are almost always significant, with the 95% credible intervals not crossing over 0 for most combinations of water quality variables and climate zones."

**Responses to Reviewer #1**

Our responses are in blue and revisions are underlined.

**Summary:**

1. Guo et al. investigate the variability in C-Q relationships in relation to the catchment hydrological conditions (more specifically the BFI) for several water quality parameters across several climate regions in Australia. The authors make use of an impressive data set from an arid region and apply a Bayesian Hierarchical Approach including the BFI, which allows understanding spatial patterns

of export dynamics. This study can thus provide an important contribution for understanding solute transport beyond temperate regions. However, the manuscript still needs more clarity on the research questions and key messages and methodological improvements. I would suggest the manuscript for major revisions as substantial improvements are still necessary.

Thank you for acknowledging the value of the study, and your comprehensive suggestions for further improvement. We provide below a point-to-point response to your comments, along with our corresponding revisions.

**General comments:**

2. One of my concerns is that given the fact that previous studies in the same region, using the same dataset and similar methods (as I read from the text), are not accessible or provided (under review or in preparation), it is not possible to judge the additional value of this study. The preceding studies (Lintern and Liu) are referenced both when defining the research goals and in the method sections. I definitely see the value of investigating C-Q relationships in various climate zones, but this was also done by these referenced studies. It is hard to judge the additional value without knowing what was shown already.

We believe this comment refers to L65-69 in the Introduction of the original manuscript, which summarized our preceding studies, Lintern et al. (2021, recently published) and Liu et al. (in preparation). Both studies focused on how C-Q relationships vary across the Australian continent. Considering the paper timelines, we decided to keep only Lintern et al. (2021) here and add a summary of this study in the revised manuscript as:

- L79: "The current knowledge gap in understanding catchment export regimes for regions other than Europe and North America was partially addressed in Lintern et al. (2021), which focused on differences in water quality status and the C-Q relationships across different climate zones in the Australian continent. One remaining question that Lintern et al. (2021) highlighted is our lack of understanding of the substantial variations in C-Q relationships within each climate zone."

We also added the following discussion on new insights added by this current study to the end of the Introduction, as:

- L99: "We use a subset of the grand dataset that Lintern et al. (2021) used, which enables us to focus on representative catchments with water quality records captured under a wider range of flow conditions."
- 3. The motivation of investigating BFI impact on C-Q relationships was not convincing for me. It needs to be clear 1.) why we need to know that and 2.) what exactly we do not know yet. The first question is not satisfyingly presented: Why do you want to focus on BFI, why is it useful to investigate this relationship? For the second: From my knowledge and in contrast of what you state (see also my comments below), the influence of BFI on the spatial variability of C-Q relationships has been discussed in several previous studies. However, I agree that studies have been biased towards temperate climates. I think the latter should be the main motivation, while generally the literature review on the control of BFI needs to be extended. It is not right, that is has not been investigated. There are studies using BFI as a descriptor for explaining the variability in export behaviour of different solutes, including several studies that you have cited in the introduction but considering other statements. For example, Minaudo et al. 2019 stated, "we found for NO3- that high BFI values, low W2, and low erosion differentiated C-Q dilution patterns from non-significant and mobilization types". But also Ebeling et al. 2021, Moatar et al. 2017, Musolff et al. 2015 have used the BFI to explain variability in C-Q relationships among catchments

for several solutes. Moatar et al. 2020 also investigated the impact of discharge flashiness on C-Q slopes and subsequently load flashiness. As BFI and Q flashiness are closely linked, this needs to be mentioned in the introduction. These also need to be discussed in relation to your study in the discussion. Also see further comments below

This study is important because understanding the impact of baseflow contribution on C-Q relationships can help explain both the spatial and temporal variation of C-Q relationships, and to infer key transport pathways.

We agree with the reviewer that previous studies have investigated how variation in baseflow contribution affects C-Q relationships across multiple catchments. The link between C-Q patterns and interannual baseflow index (or similar metrics) were well studied (e.g., Moatar et al. 2017, 2020, Musolff et al. 2015, Ebeling et al. 2021), while Minaudo et al. (2019) explicitly explored the link between instantaneous BFI and C-Q relationships. However, the existing studies on the impact of baseflow contribution are largely focused on the temperate climate. Therefore, we haven't yet well understood the effects of baseflow contribution across a wider range of climate zones, nor been able to quantify this impact.

The key motivation of this study is to understand and quantify how catchment-level metrics of baseflow contribution affect C-Q relationships across catchments in a wide range of climate zones. Through synthesizing continental water quality data, we were able to gain large-scale understanding of how C-Q changes in catchments with different baseflow conditions.

Our specific revisions to clarify these include:

- 1) Added the following sentences to the end of the Introduction to highlight the importance of the key issue that this study explores:
- L101: "By analysing the impacts of baseflow contributions on C-Q relationships, this study will i) explain the variations in C-Q relationships within individual climate zones; ii) broaden the existing knowledge of how baseflow contribution impacts C-Q relationships to a wider range of climate conditions, and thus infer key constituent transport pathways in different climate zones."
  - 2) Revised the literature review and add discussions on the state-of-art research on the impacts of baseflow contribution in the Introduction:
- L59: "Prior studies have explored the links between C-Q relationships and baseflow index (BFI) and similar hydrological metrics at an interannual scale (e.g., Ebeling et al., 2021; Moatar et al., 2017; Musolff et al., 2015) or at the scale of storm events (e.g., Knapp et al., 2020; Minaudo et al., 2019; Musolff et al., 2021). Across both long and short timescales, a consistent finding is that, within a particular catchment, the C-Q relationship (and thus export behaviour) is dependent on whether streamflow is dominated by baseflow or quickflow, i.e., the baseflow contribution to total flow (Gorski & Zimmer, 2021; Knapp et al., 2020; Minaudo et al., 2019). These studies also identified baseflow contribution as a key driver of the variation in C-Q relationships across catchments (Musolff et al., 2015; Moatar et al., 2017). For example, Knapp et al. (2020) found that for solutes that are partly derived from atmospheric inputs, such as nitrate and chloride, mobilisation behaviours (i.e., positive C-Q slopes) often occur during events with drier antecedent conditions. For nitrate, baseflow contributions can further affect the C-Q relationships via changing the connectivity between surface flow and groundwater (Minaudo et al., 2019). Baseflow variation also affects the capacity of nutrient removal via changing the relative importance of hydrological and biogeochemical processes (Moatar et al., 2017). Further, the variation in the baseflow contribution of a catchment is also a key feature that can be linked to the shift between different

dominant flow paths during low- and high-flow (e.g., von Freyberg et al., 2018), leading to contrasting sources and mobilisation behaviours for solutes and particulates."

- 3) Improved clarification of the knowledge gap on the lack of understanding of how C-Q relationships vary with baseflow contribution across a wide range of climate zones.
- L73: "Although a substantial body of knowledge has been established on the impact of baseflow contributions on C-Q relationships, the existing studies have largely focused on catchments in temperate climates in Europe and North America (Knapp et al., 2020; Gorski & Zimmer, 2021; Minaudo et al., 2019; Musolff et al., 2015). The narrow range of climate conditions explored so far implies a potential limitation in transferring and systematically comparing new findings to other climate zones and other parts of the world, because climate is proven a key control of the hydrological regime, especially regarding the baseflow contribution and flow paths of individual catchments (Beck et al., 2013; von Freyberg et al., 2018)."
- 4.
- a) Some of the methods seem inappropriate, especially as there is too few data for some climate-solute combinations to fit robust models/regressions and interpret them (further comments below).
- b) Besides, I do not see the value of investigating the BFI impact within each climate zone individually, i.e. separating the climate zones and fitting different models, instead of investigating the BFI impact across the whole climate variability. I think it would be more valuable to know what effect the BFI has across the whole climatic variability, i.e. the continuum of variations. The climate zones, could be represented by their characteristics such as precipitation amount, seasonality, aridity, temperature etc.. Even within the climate zones those variables vary and could potentially explain the deviations not explained by BFI.

**a)**

In this study, we used a single Bayesian Hierarchical Model (BHM) to fit all data points for each constituent simultaneously, across all climate zones. Although we used different parameters for individual climate zones, this model structure has a great advantage of 'borrowing power', meaning that information is shared across climate zones, and parameter fitting for one climate zone can utilize data from all other climate zones. As illustrated in Figures 5 and 6 (both in the original and revised manuscripts), BHM has been effective in estimating the impact of baseflow contribution (summarized by catchment median BFI and range of daily BFI) on C-Q slopes even for climate zones with limited data, while acknowledging the model uncertainty. This model enables us to estimate the BFI impacts for climate zones with limited data by making use of data from other climate zones; in this way we made the best use of all available data.

To address this comment, we added further discussion to both the Introduction and the Method Section 2.2 (which introduces the modelling approach) on the key advantages of BHM in effectively handling data-limited situations and spatio-temporal data with uneven coverage, and justifications for why BHM is suited to the dataset analysed in this study, as:

- L96: "We answer our research questions and test our hypotheses with a Bayesian hierarchical model (BHM) (Gelman et al., 2013), which is an integrated framework that enables sharing information across catchments to strengthen the statistical power of explaining variation in individual catchments. The model is a powerful approach to capture water quality variability across catchments of varying conditions and record lengths, which is the case for Australian water quality data (Guo et al., 2019, 2020; Liu et al., 2021)." L168: "The key reason for choosing this model is the high heterogeneity in the national C-Q dataset in both the record period and the representation of individual climate zones, as illustrated in Section 2.1.1. BHM is effective in handling data-limited situations via its 'information sharing' or 'borrowing power' across space (Gelman et al., 2013; Webb & King, 2009), which has been shown to be highly effective in explaining variability in spatial-temporal data under data-limited situations. This has been highlighted in several recent studies in modelling water quality over large regions in Australia (Guo et al., 2019, 2020; Liu et al., 2021). Another advantage of BHM is the ability to account for uncertainty, which is especially important for analysing water quality data, as these data are often associated with high uncertainty due to sparse sampling of the natural variability of chemical species in river flow (Guo et al., 2020; Liu et al., 2021)."

The following references were added to the manuscript support the above arguments:

- Guo, D., Lintern, A., Webb, J. A., Ryu, D., Bende-Michl, U., Liu, S. & Western, A. W. (2020). A databased predictive model for spatiotemporal variability in stream water quality. Hydrology and Earth System Sciences, 24(2), pp. 827-847.doi:10.5194/hess-24-827-2020
- Guo, D., Lintern, A., Webb, J. A., Ryu, D., Liu, S., Bende-Michl, U., . . . Western, A. W. (2019). Key Factors Affecting Temporal Variability in Stream Water Quality. Water Resources Research, 55(1), 112-129. doi:10.1029/2018wr023370
- Liu, S., Ryu, D., Webb, J. A., Lintern, A., Guo, D., Waters, D., & Western, A. W. (2021). A Bayesian approach to understanding the key factors influencing temporal variability in stream water quality
   a case study in the Great Barrier Reef catchments. Hydrol. Earth Syst. Sci., 25(5), 2663-2683. doi:10.5194/hess-25-2663-2021

**b)**

The reason to investigate BFI effects for individual climate zones separately is to see whether we could identify any statistically significant differences of the effects for different climate zones, and the results suggested that this is the case (from the C-Q slopes in Figure 5 and the simulated C-Q~BFI relationships in Figure 6, in both the original and revised manuscripts). If there is no significant difference of the BFI effects between climate zones, the model is also capable of indicating this – as would be shown with similar, undistinguishable modelled effects of BFI for individual climate zones. Our current model structure enables us to test this hypothesis, since we are essentially developing different models for each climate zone and comparing them within one larger modelling framework. In contrast, a model with only a single parameter for the BFI effects across all climate zones cannot be used to test this hypothesis. We added the following sentences to clarify this in Method Section 2.2, which introduces the modelling approach to exploring the BFI effects:

- L189: "We chose to investigate the effects of baseflow contributions for individual climate zones separately to identify any statistically significant differences of these impacts between climate zones. If there is no significant difference between climate zones, the model is also capable of indicating this – as would be shown with similar, undistinguishable modelled effects of baseflow contribution for individual climate zones. Thus, our BHM incorporates different models for individual climate zones and compares them within one comprehensive modelling framework."

To illustrate the value added by considering climate-specific effects of BFI, we replaced the previous results in Section 3.3.1 (now as part of Section 3.3) with a new comparison of performance between the lumped and climate-specific models. The key results added are included in the revised Table 1 as below:

Table 1. Performance of the BFI-based C-Q models – the columns show four alternative model structures with *BFI\_m* or *BFI\_range* as the key predictor, and with the impacts of baseflow contribution considered as lumped or specific to individual climate zones. The rows show results for individual water quality parameters. All model performances are summarised by R2, which quantifies the percentage of variance in C-Q slopes explained by the BFI-based models.

| WQ
paramete | Media
n C-Q | Current (climate- | Current (climate-specific impacts) |       | Baseline (lumped impact across climate zones) |  |
|----------------|----------------|-------------------|------------------------------------|-------|-----------------------------------------------|--|
| r              | slope          | BFI_m             | BFI_range                          | BFI_m | BFI_range                                     |  |
| TSS            | 0.15           | 0.16              | 0.11                               | 0     | 0.04                                          |  |
| ТР             | 0.09           | 0.14              | 0.17                               | 0     | 0.08                                          |  |
| SRP            | 0.06           | 0.02              | 0                                  | 0.03  | 0.05                                          |  |
| TN             | 0.09           | 0.18              | 0.12                               | 0.02  | 0.03                                          |  |
| NOx            | 0.36           | 0.22              | 0.18                               | 0.03  | 0                                             |  |
| EC             | -0.07          | 0                 | 0.01                               | 0     | 0.01                                          |  |

We added new interpretation of these new results as:

L305: "Using catchment-level metrics of baseflow contribution alone (either BFI m or BFI range) can explain up to 22% of the variation in catchment C-Q slopes. Although these results represent limited model predictive capacity, the model does cover a large range of catchment conditions such as contrasting land uses and hydro-climate conditions. Therefore, the amount of variation that can be explained by a single BFI metric highlights baseflow contribution as an important factor that influences catchment C-Q relationships. Further, it is also worth highlighting that incorporating climate-specific impacts of baseflow contribution is highly beneficial in explaining these variations. For all six water quality parameters, the baseline model – which uses a lumped effect of catchment baseflow contribution across different climate zones – can barely explain any variation in the C-Q slopes (with all  $R^2 < 0.08$ , i.e., <8% of the variation explained). In contrast, the climate-specific models generally offer up to 20% increase in the variance explained for C-Q slopes, except for EC and SRP, for which performance is equally low regardless of whether the effects of baseflow contribution are separated for individual climates. The low performances for EC and SRP are likely attributed to the smaller magnitudes of C-Q slopes as highlighted in the lower median *C*-*Q* slope in Table 1, making it statistically more difficult to explain variations across catchments for these two water quality variables. These results further emphasise that in general, the impacts of catchment baseflow contribution on C-Q slopes are better defined within individual climate zones, which confirms the validity of our BFI-based C-Q models (Eqn. 3)."

We expect that the abovementioned revision also removes the previous confusion as highlighted in your Comments #24 and #26 (see details in our specific responses to those comments).

We agree with you that variations in the C-Q slopes can also be explored together with other climate drivers (e.g., temperature, rainfall, aridity), but a key challenge we see in other water quality modelling studies with these potential drivers is the high cross-correlation between these variables, thus we decided to use climate zones as an integrator of many hydroclimatic variables. The focus of this study is BFI and we hypothesise that BFI has a significant effect on C-Q slopes – which is supported by the study results.

To address this comment, we have added further justification to the use of climate zones in Section 2.2.

- L189: "We chose to investigate the effects of baseflow contributions for individual climate zones separately to identify any statistically significant differences of these impacts between climate zones. If there is no significant difference between climate zones, the model is also capable of indicating this – as would be shown with similar, undistinguishable modelled effects of baseflow contribution for individual climate zones. Thus, our BHM incorporates different models for individual climate zones and compares them within one comprehensive modelling framework."

We also added discussions on the need for further studies on the impact of individual climate drivers on C-Q slopes in the Conclusion.

- L463: "This study used catchment-level metrics of baseflow contribution as the only predictor of C-Q slopes. The baseflow contribution alone can explain up to 22% variance in the C-Q slopes across the Australian continent. This highlights a substantial role in baseflow contribution in shaping the C-Q relationships, while also suggesting the need of further work to synthesise the impacts of baseflow contribution together with other spatial drivers (e.g., climate, land use, land cover and geology) to include their interactions and establishing their relative importance on influencing C-Q relationships."
- 5. The interpretation of BFI\_m in terms of variability of flow paths is not convincing to me as you could easily and more directly use the range of BFI to determine the relationships between C-Q slope variability with BFI ranges. I think it would be good to look at this instead of speculating, as you have the data at hand and Figure 3b is not convincing enough for this interpretation, in my opinion. Instead of the range BFI\_h-BFI\_I you could also consider other metrics of variability.

To address this comment, we have added substantial new analyses to explore the impact of catchment BFI range on the C-Q slopes. Specifically, we explored a new model with the same structure as our original model based on *BFI\_m*, which is now based on *BFI\_range* (i.e. the difference between the 10th and 90th quantiles of daily BFI). We revised Figures 5 and 6 to include results from both models, which illustrated that the effects of *BFI\_range* on C-Q slopes are highly consistent with the modelled effects of *BFI\_m* on C-Q slopes. These new results provide more concrete evidence on our previous 'speculation' on how the BFI range in a catchment can impact its C-Q slope.

According to these new results, we have substantially revised the discussions of these results which are presented after Figures 6 and 7 in Section 3.3.

Figure 5. Modelled effects of *BFI\_m* and *BFI\_range* on catchment C-Q slopes for each climate zone ( $\delta_{BFI_climate}$ ) for each water quality parameter. The bars show the 95% credible intervals (the range between 2.5th to 97.5th percentiles of Bayesian posterior distribution) of the modelled effects, and the dots indicate the corresponding median levels. The colours indicate whether an effect is significantly positive (red), significantly negative (blue), or non-significant (grey); a positive effect means that the C-Q slope increases with a higher catchment *BFI\_m* or *BFI\_range*, and vice versa. Black

---

## Author Response (AR2)

**Response to Review Comments (hess-2021-353)**

Nov 23 2021

**Editor's comment**

Please remove the "the black dashed lines show the reference 1:1 line" in Figure 4. You are plotting log(C) versus log(Q), the 1:1 line has no meaning.

**Authors' response**

Thank you. We have revised Figure 4 as suggested, with all 1:1 lines removed. We also included the high-resolution versions of all figures along with the corrected manuscript.